# LMDX: LANGUAGE MODEL-BASED DOCUMENT INFORMATION EXTRACTION AND LOCALIZATION

## ABSTRACT

Large Language Models (LLM) have revolutionized Natural Language Processing (NLP), improving state-of-the-art on many existing tasks and exhibiting emergent capabilities. However, LLMs have not yet been successfully applied on semi-structured document information extraction, which is at the core of many document processing workflows and consists of extracting key entities from a visually rich document (VRD) given a predefined target schema. The main obstacles to LLM adoption in that task have been the absence of layout encoding within LLMs, critical for a high quality extraction, and the lack of a grounding mechanism ensuring the answer is not hallucinated. In this paper, we introduce *Language Model-based Document Information EXtraction and Localization* (LMDX), a methodology to adapt arbitrary LLMs for document information extraction. LMDX can do extraction of singular, repeated, and hierarchical entities, both with and without training data, while providing grounding guarantees and localizing the entities within the document. Finally, we apply LMDX to the PaLM 2-Small LLM and evaluate it on VRDU and CORD benchmarks, setting a new state-of-the-art and showing how LMDX enables the creation of high quality, data-efficient parsers.

## 1 INTRODUCTION

The recent advent of transformers (Vaswani et al., 2017) and self-supervised pretraining procedures has led to significant progress in Visually Rich Document (VRD) Understanding. Within that field, the task of document information extraction (IE), which consists of extracting key entities within a semi-structured document (e.g. invoice, tax form, paystub, receipt, etc) given a predefined schema, has received a lot of attention from industry and academia due to its importance and wide applicability to intelligent document processing workflows. However, document information extraction still remains challenging for current generation systems. In particular, information in semi-structured forms is organized in complex layout across many possible templates, which requires understanding of the document context, spatial alignment among the different segments of text, and tabular arrangement of structured entities (e.g. line items on an invoice, deduction items on a paystub, etc.). Content on the document can be printed or handwritten, with scanning artefacts like rotation and contrast issues. Moreover, since some business automation workflows require certain level of accuracy, they are often integrated with human-in-the-loop interactions for auditing and correction of predictions, requiring knowing the precise location of extracted entities to make it a tractable task for a human rater. Finally, since a quasi-infinite number of document types exist, and that organizations have limited annotation resources, most parsers are built with very small amount of data.

From those complexities emerge the following desiderata of document information extraction systems: they should support high-quality extraction of singular, repeated, and hierarchical entities, while localizing those entities in the document, and doing so with very low or no amount of training data. So far, no publicly disclosed system has been able to address all of those desiderata.

Many current approaches divide the problem in two stages: a text recognition/serialization step, usually achieved by an off-the-shelf Optical Character Recognition (OCR) service, followed by a parsing step, which finds the relevant entity values from the recognized text. Since the text serialization is imperfect, much attention has been given to fusing the text and layout together in the parsing step (Majumder et al., 2020; Garncarek et al., 2021; Hwang et al., 2021; Katti et al., 2018; Denk & Reisswig, 2019). Hong et al. (2021) proposes to encode the relative 2D distances of text blocks in

the attention of the transformer, and learning from unlabeled documents with an area-masking strategy. Lee et al. (2022) proposes encoding the relative token positions with a graph neural network with edges constructed from a beta-skeleton algorithm. It further frames information extraction as a Named Entity Recognition (NER) task with an Inside-Outside-Begin (IOB) token tagging scheme (Ramshaw & Marcus, 1995; Palm et al., 2017) which allows them to localize the entities. However, IOB does not support extracting hierarchical entities, and is not robust to text serialization errors, where an entity is broken in disjoint segments.

Since text and layout do not contain all the information in the document (e.g. table boundaries, logos), leveraging the image modality has also been extensively explored (Xu et al., 2021; Lee et al., 2023; Appalaraju et al., 2021; 2023; Zhang et al., 2022). Xu et al. (2020) uses a separate image encoder before adding the output as feature to the token encodings, while Huang et al. (2022) jointly models the page image patches alongside the tokens, using a word-patch alignment self-supervised pretraining task to learn the connection between the modalities.

Other approaches treat extraction as a sequence generation problem. Powalski et al. (2021) adds an auto-regressive decoder on top of a text-layout-image encoder, all initialized from T5 (Raffel et al., 2020). Kim et al. (2022) foregoes the text recognition step completely, using a Vision Transformer encoder with an auto-regressive decoder pretrained on a pseudo-OCR task on a large document image corpora, and finetuned on the final extraction parse tree with Extensible Markup Language (XML) tags for the target extraction schema. While this approach allows to predict hierarchical entities, it does not allow localizing entities in the document.

None of the previously discussed approaches attempt to understand the semantics of the schema and its entity types, and instead opt to encode the schema in the model weights through training, hence requiring training data for unseen schemas and document types. QueryForm (Wang et al., 2023b) utilizes a prompt encoding both the schema and entity types, allowing the model to do zero-shot extraction. Likewise, Wei et al. (2023) inputs the raw entity types in the encoder itself, and uses a scoring matrix to predict the link classes between document tokens and types, with great few-shot performance. However, both approaches are not able to predict hierarchical entities.

In parallel, Large Language Models (OpenAI, 2023; Google et al., 2023; Hoffmann et al., 2022) have revolutionized Natural Language Processing, showing the capabilities to solve a task with simply an instruction (Wei et al., 2022) or a few examples added to the prompt (Brown et al., 2020). This paradigm opens the possibility of extracting entities with very little to no training data. Wang et al. (2023a) transforms the NER task to a sequence generation task suitable for LLMs by incorporating special tokens in the sequence, marking the entity boundaries, and proposes a self-verification strategy limiting the LLM hallucinations. However, this is applicable to text-only scenarios, with hallucinations still a possibility.

This motivates us to introduce *Language Model-based Document Information EXtraction and Localization* (LMDX), a methodology for leveraging off-the-shelf LLMs for information extraction and localization on semi-structured documents. Our contributions can be summarized as follows:

- We propose a prompt that enables LLMs to perform the document IE task on leaf and hierarchical entities with precise localization, including without any training data, and using only the simple text-in, text-out interface that all LLMs provide.

- We also propose a layout encoding scheme that communicate spatial information to the LLM without any change to its architecture.

- We introduce a decoding algorithm transforming the LLM responses into extracted entities and their bounding boxes on the document, while discarding all hallucination.

- We systematically evaluate the data efficiency of LMDX on multiple public benchmarks and establish a new state-of-the-art on those by a wide margin, especially at low-data regimes.

A comparison of LMDX characteristics and other popular document information extraction systems can be found at Table 1.

Table 1: Comparison of document information extraction systems.

| Document Information Extraction Systems | Hierarchical entity | Entity localization | Zero-shot support |
|---|:---:|:---:|:---:|
| FormNet(v2), LayoutLM(v2), Docformer, Glean, ... | ✗ | ✓ | ✗ |
| QueryForm, PPN | ✗ | ✓ | ✓ |
| Donut | ✓ | ✗ | ✗ |
| **LMDX (Ours)** | ✓ | ✓ | ✓ |

## 2 METHODOLOGY

### 2.1 OVERVIEW

Overall, our pipeline is divided into five stages: OCR, chunking, prompt generation, LLM inference and decoding, detailed in the following sections. An overview with a simple example can be found in Figure 1, with the input and output of each stage showcased. In this example, the target schema contains two leaf entity types *retailer* and *subtotal*, and one hierarchical entity type *line_item*, composed of a *product_id* and a *product_price*.

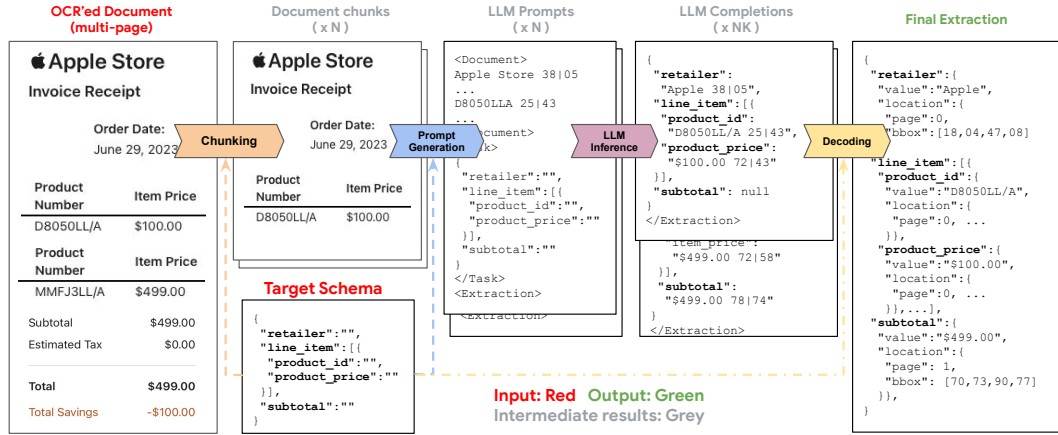

Figure 1: Overview of the LMDX methodology.

### 2.2 OPTICAL CHARACTER RECOGNITION

We first use an off-the-shelf OCR service on the document image to obtain words and lines segments, along with their corresponding spatial position (bounding box) on the document. An example of output from that stage on a sample document is given in Appendix A.6.

### 2.3 CHUNKING

While some LLMs support long context (hundreds of thousands of tokens), not all LLMs can fit the entire document within its prompt. For those cases, the document is divided into document chunks so that each is small enough to be processed by the LLM. To achieve this, we first divide the document into individual pages, then we iteratively remove the last line segments (coming from OCR) until the prompt containing this chunk is below the maximum input token length of the LLM. Lastly, we group those removed lines as a new document page, and repeat the same logic until all chunks are below the input token limit of the LLM. At the end of this optional stage, we have $N$ chunks. The decision to first divide the document by page stems from the observation that entities rarely cross page boundaries, and as such this chunking scheme will have minimal impact on the final extraction quality. The algorithm is described in pseudo-code in Appendix A.1.

## 2.4 PROMPT GENERATION

The prompt generation stage takes in the $N$ document chunks and creates a LLM prompt for each of them. As seen in Figure 2, our prompt design contains the document representation, a description of the task, and the target schema representation containing the entities to extract. XML-like tags are used to define the start and end of each component.

```
<Document>
{DOCUMENT_REPRESENTATION}
</Document>
<Task>
{TASK_DESCRIPTION}
{SCHEMA_REPRESENTATION}
</Task>
<Extraction>
```

Figure 2: Structure of the LLM prompts.

**Document Representation.** The chunk content is represented in the prompt as the concatenation of all its segment texts, suffixed with the coordinates of those segments in the following format: $< segment\ text > \ XX|YY_{segment}$. Coordinate tokens are built by normalizing the segment's X and Y coordinates, and quantizing them in $B$ buckets, assigning the index of that bucket as the token for a coordinate.

This coordinate-as-tokens scheme allows us to communicate the layout modality to the LLM, without any change to its architecture. There are many variation to that scheme: using OCR line versus OCR words as segment, the granularity of the quantization, and the number of coordinates to use per segment (e.g. $[x_{center}, y_{center}]$ versus $[x_{min}, y_{min}, x_{max}, y_{max}]$). Appendix A.4 shows how those variations affect the prompt token length. In all our experiments, we use line-level segments with 2 coordinates $[x_{center}, y_{center}]$ and $B = 100$ quantization buckets.

**Task Description.** The task description is simply a short explanation of the task to accomplish. In our experiments, we hard code it to the following: *From the document, extract the text values and tags of the following entities:*.

**Schema Representation.** The schema is represented as a structured JSON object, where the keys are the entity types to be extracted, and the values correspond to their occurrence (single or multiple) and sub-entities (for hierarchical entities). For instance, *{"foo": "", "bar": [{"baz": []}]}* means that the LLM should extract only a single entity of type *foo* and multiple hierarchical entities of type *bar*, that could each hold multiple entities of type *baz*.

After this step, we have $N$ prompts, one for each document chunk. A full example of a prompt on a document can be found in Appendix A.6.

## 2.5 COMPLETION TARGETS

In this section, we describe the expected LLM completion format, which can be observed in Figure 1. Like the schema, the completion is a JSON structured object with the keys being the entity types, and values being the extracted information from the document chunk. JSON was chosen as a format for the completion and schema since it supports hierarchical objects, is very token-efficient, and usually present in LLMs training data mixtures. Note that the keys in the completion have the same ordering, occurrence and class (hierarchical or leaf) as the entity types in the schema. The values of leaf entities must follow a specific format:

$$< text\ on\ segment_1 > \ XX|YY_{segment_1} \backslash n < text\ on\ segment_2 > \ XX|YY_{segment_2} \backslash n \ ...$$

An entity can span multiple (potentially disjoint) OCR segments (lines or words). For each segment of the entity, the value contains the entity text on that segment, along with the coordinate tokens of that segment, which act as a *segment identifier*, allowing us to localize the entities and ground the model prediction (e.g. making sure the extracted value is not a hallucination), as will be detailed in

Section 2.7. Finally, missing entity types are completed by the model with $null$ for singular types, and [] for repeated types. Samples of completions can be found in Appendix A.6.

## 2.6 LLM INFERENCE

In this stage of the pipeline, we run inference on the LLM with the $N$ prompts. For each prompt, we sample $K$ completions from the LLM (for a total of $NK$ completions for the entire document) using $\text{Top}_K$ sampling. This randomness in the sampling allows to do error correction (e.g. if a response is not valid JSON, have hallucinated segment coordinate identifier, etc), and increase the extraction quality as will be shown in further sections. Note that we still want the inference to be fully deterministic so that LMDX's extractions are the same across two identical documents. To do so, we rely on pseudo-random sampling using a fixed seed.

## 2.7 DECODING

In this stage, we parse the raw LLM completions into structured entities and their locations.

**Conversion to structured entities.** We begin by parsing each model completion as a JSON object. Completions that fail to parse are discarded. For each key-value pair in the JSON object, we interpret the key as the entity type and parse the value to get the entity text and bounding box (as detailed in the next paragraph). Predicted entity types that are not in the schema are discarded. If the model unexpectedly predicts multiple values for single-occurrence entity types, we use the most frequent value as the final predicted value. Hierarchical JSON object are recursively parsed as hierarchical entities in a similar manner. This algorithm is described in pseudo-code in Appendix A.3.

**Entity Value Parsing.** We expect the JSON value to include both text extractions and segment identifiers for each predicted entity, as described in Section 2.5. We first parse the value into its $(segment\ text, segment\ identifier)$ pairs. For each pair, we look up the corresponding segment in the original document using the segment identifier and verify that the extracted text is exactly included on that segment. The entity is discarded if that verification fails, ensuring LMDX discards all LLM hallucinations. Finally, once we have the entity location on all its segments, we get the entity bounding box by computing the smallest bounding box encompassing all the OCR words included in the entity. Entity values with any segments that fail to ground (invalid entity value format, non-existent segment identifier, or non-matching segment text) in the original document are discarded. The entity value parsing algorithm is described in pseudo-code in Appendix A.2, and parsing errors rates are detailed in Appendix A.9.

**Prediction Merging.** We first merge the predicted entities for the same document chunk from the $K$ LLM completions through majority voting (Wang et al., 2022). For each entity type, we gather the predicted entities, including empty predictions, across the $K$ completions. The most common prediction(s) are selected as the predicted value for that entity type. We then merge the predictions among the $N$ document chunks by concatenating them to obtain the document level predictions.

**Prediction Merging for hierarchical entities.** For hierarchical entities, we use the entire predicted tree value from a single LLM completion, as this method best preserves the parent-child relationship predicted by the model. For each top-level hierarchical entity type, we perform majority voting on all affiliated leaf, intermediate and top-level entity types among $K$ completions as if they are flattened. We then tally the votes with equal weight to determine which completion to use for the prediction, and select the most common one for that hierarchical entity.

## 3 EVALUATION

We evaluate the methodology from section 2 on public benchmarks using the PaLM 2-Small LLM, which we call LMDX$_{\text{PaLM 2-Small}}$. Note that we use the small version of this LLM due to limited accelerator resources, but larger versions could be used, likely leading to higher extraction quality.

Our training process is composed of two phases. In the first phase we finetune PaLM 2-Small on a data mixture containing a variety of *(document, schema, extraction)* tuples. In particular, this data

mixture contains the *Payment* dataset (Majumder et al., 2020), along with a diverse set of publicly available PDF form templates obtained from government websites that we filled with synthetic data using an internal tool, and annotated for schema and entities to extract. The goal of this phase is to obtain a Base Entity Extractor model by training the model to interpret the semantics of the entity types and extraction hierarchy specified in the schema, and find them within the document, along with learning the extraction syntax. Hence, the variety of schemas and documents in this phase is of utmost importance. This model is used for doing zero-shot extraction on a wide variety of document types.

During the second phase, starting from the base entity extractor checkpoint from the previous phase, we finetune the LLM on the target to specialize it to do high quality extraction on the target benchmark. At this stage, only the target benchmark data is included in the training mixture. Note that, for zero-shot experiments, this second phase is skipped. Furthermore, no document or schema contained in the base extraction training phase overlap with the documents and schemas used in the specialization training phase. For all training phases, we follow the input and target syntax described in section 2.4 and 2.5.

### 3.1 PARAMETERS

For training, we finetune PaLM 2-Small using a batch size of 8, a dropout probability of 0.1 and a learning rate of $10^{-6}$ with a standard cross-entropy loss. Once training is done, we select the checkpoint with the lowest loss on the dev set, and report performance on the test set. For LLM inference, we use a temperature of 0.5 and a $Top_K$ of 40, sampling 16 responses for each chunk processed by the LLM, as described in section 2.6. Finally, for both training and inference, we use an input token length of 6144 and output token length of 2048. We use line-level segments and only two coordinates [$x_{center}$, $y_{center}$] with 100 quantization buckets to save on the number of input and output tokens consumed by the coordinate-as-tokens scheme, as supported by Appendix A.4.

### 3.2 DATASETS

**Visually Rich Document Understanding (VRDU).** Wang et al. (2023c) introduces a public benchmark for entity extraction from visually-rich documents that includes two datasets: Registration Form, containing 6 semantically rich entity types, and Ad-buy Form, containing 14 entity types with one hierarchical *line_item* entity. For each dataset, VRDU proposes samples of 10, 50, 100 and 200 train documents to evaluate the data efficiency of models. It also offers different tasks to evaluate the generalization powers of extraction systems: Single Template Learning (STL) where train/test share the same single template, Unseen Template Learning (UTL) where train/test contain disjoint sets of templates, and Mixed Template Learning (MTL) where train/test contain overlapping sets of templates. For our experiments, we finetune LMDX$_{PaLM 2-Small}$ for 4000 steps on each dataset, training data size, and task setup independently and report Micro-F1 through the provided evaluation tool. We then compare LMDX$_{PaLM 2-Small}$ to its published state-of-the-art baselines.

**Consolidated Receipt Dataset (CORD).** Park et al. (2019) introduces a benchmark of Indonesian receipts from shops and restaurants, with a target schema of 30 fine-grained entities, grouped into *menu*, *total* and *subtotal* hierarchical entities. CORD[1] does not provide a standard evaluation toolkit, so we adopt the normalized Tree Edit Distance accuracy (n-TED) metric (Zhang & Shasha, 1989), previously introduced by Kim et al. (2022) on that benchmark, since it is agnostic to the output scheme used and considers the hierarchical entities as part of the metric. For our experiments, we use the official $800train/100dev/100test$ split, but also sample the first $D = 10/50/100/200$ documents from the train split to assess the data efficiency of LMDX on this benchmark. For each data setup, we finetune LMDX for 12000 steps. For comparison, we also train and evaluate state-of-the-art baselines $LayoutLMv3_{LARGE}$ and $Donut$. Those baselines are detailed in Appendix A.8.

**For all benchmarks.** We use the publicly provided OCR for the LMDX model and baselines with text input, ensuring a fair comparison between them. Furthermore, we also compare LMDX$_{PaLM 2-Small}$ to large model baselines on all benchmarks in the zero-shot ($|\mathcal{D}| = 0$) setting: GPT-3.5 that we prompt with the raw OCR and extraction instruction, and LLaVA-v1.5-13B that we

---

[1]https://huggingface.co/datasets/naver-clova-ix/cord-v1

prompt with the document image and extraction instructions. Those baselines are fully detailed in Appendix A.7. Unlike LMDX, those large model baselines do not localize their predictions.

## 3.3 RESULTS

Table 2: Results of LMDX$_{\text{PaLM 2-Small}}$ on the different tasks and training data size setups $|\mathcal{D}|$ of VRDU, with best performing model results in bold.

| $|\mathcal{D}|$ | Model | Localization | Registration Form | | | | Ad-buy Form | | | |
|---|---|---|---|---|---|---|---|---|---|---|
| | | | Single | Unseen | Mixed Template | | Unseen | Mixed Template | | |
| | | | Micro-F1 | Micro-F1 | Micro-F1 | Localization Accuracy | Micro-F1 | Micro-F1 | Line Item F1 (Hierarchical) | Localization Accuracy |
| 0 | LLaVA-v1.5-13B | ✗ | 5.29 | 5.05 | 5.00 | N/A | 0.38 | 0.34 | 0.00 | N/A |
| | GPT-3.5 | ✗ | 67.23 | 67.49 | 63.86 | N/A | 29.84 | 30.05 | 7.65 | N/A |
| | LMDX$_{\text{PaLM 2-Small}}$ | ✓ | **73.81** | **74.94** | **71.65** | **93.21** | **39.33** | **39.74** | **21.21** | **88.18** |
| 10 | FormNet | ✓ | 74.22 | 50.53 | 63.61 | - | 20.28 | 20.47 | 5.72 | - |
| | LayoutLM | ✓ | 65.91 | 25.54 | 36.41 | 98.71 | 19.92 | 20.20 | 6.95 | 92.60 |
| | LayoutLMv2 | ✓ | 80.05 | 54.21 | 69.44 | 99.00 | 25.17 | 25.36 | 9.96 | 93.95 |
| | LayoutLMv3 | ✓ | 72.51 | 21.17 | 60.72 | 99.20 | 10.01 | 10.16 | 5.92 | 90.68 |
| | LMDX$_{\text{PaLM 2-Small}}$ | ✓ | **90.88** | **86.87** | **87.72** | **99.75** | **54.82** | **54.35** | **39.35** | **94.51** |
| 50 | FormNet | ✓ | 89.38 | 68.29 | 85.38 | - | 39.52 | 40.68 | 19.06 | - |
| | LayoutLM | ✓ | 86.21 | 55.86 | 80.15 | 99.69 | 38.42 | 39.76 | 19.50 | 95.24 |
| | LayoutLMv2 | ✓ | 88.68 | 61.36 | 84.13 | 99.54 | 41.59 | 42.23 | 20.98 | 95.64 |
| | LayoutLMv3 | ✓ | 87.24 | 47.85 | 81.36 | 99.39 | 38.43 | 39.49 | 19.53 | 95.28 |
| | LMDX$_{\text{PaLM 2-Small}}$ | ✓ | **93.06** | **88.43** | **91.42** | **99.87** | **75.70** | **75.08** | **65.42** | **98.28** |
| 100 | FormNet | ✓ | 90.91 | 72.58 | 88.13 | - | 39.88 | 40.38 | 18.80 | - |
| | LayoutLM | ✓ | 88.70 | 63.68 | 86.02 | 99.63 | 41.46 | 42.38 | 21.26 | 95.09 |
| | LayoutLMv2 | ✓ | 90.45 | 65.96 | 88.36 | 99.72 | 44.35 | 44.97 | 23.52 | 95.72 |
| | LayoutLMv3 | ✓ | 89.23 | 57.69 | 87.32 | 99.72 | 41.54 | 42.63 | 22.08 | 95.88 |
| | LMDX$_{\text{PaLM 2-Small}}$ | ✓ | **93.97** | **89.70** | **92.41** | **99.92** | **75.99** | **78.05** | **69.77** | **98.69** |
| 200 | FormNet | ✓ | 92.12 | 77.29 | 90.51 | - | 42.87 | 43.23 | 21.86 | - |
| | LayoutLM | ✓ | 90.47 | 70.47 | 87.94 | 99.69 | 44.18 | 44.66 | 23.90 | 95.38 |
| | LayoutLMv2 | ✓ | 91.41 | 72.03 | 89.19 | 99.75 | 46.31 | 46.54 | 25.46 | 95.78 |
| | LayoutLMv3 | ✓ | 90.89 | 62.58 | 89.77 | 99.67 | 44.43 | 45.16 | 24.51 | 95.95 |
| | LMDX$_{\text{PaLM 2-Small}}$ | ✓ | **93.97** | **90.22** | **92.78** | **99.87** | **78.42** | **79.82** | **72.09** | **98.65** |

Results for VRDU are presented in Table 2. For all data regimes and tasks, LMDX$_{\text{PaLM 2-Small}}$ sets a new state-of-the-art by a wide margin. In particular, we find that LMDX$_{\text{PaLM 2-Small}}$ has higher extraction quality than GPT-3.5 and LLaVA-v1.5-13B while also localizing its predictions. LMDX$_{\text{PaLM 2-Small}}$ also exhibits similar extraction quality at zero-shot than baselines at 10-100 train dataset size (for instance 39.74% Micro-F1 on Ad-Buy Form Mixed Template vs 40.68% for Form-Net at 50 train documents, or 73.81% Micro-F1 on Registration Single Template vs 74.22% for FormNet at 10 train documents). Moreover, LMDX$_{\text{PaLM 2-Small}}$ is much more data efficient than the baselines: it is at 5.06% Micro-F1 of its peak performance at 10 training documents for Registration Form Mixed Template (87.72% vs 92.78% Micro-F1) while LayoutLMv2, the strongest finetuned baseline, is within 19.75% of its peak performance (69.44% vs 89.19% Micro-F1). Lastly, LMDX$_{\text{PaLM 2-Small}}$ generalizes better to unseen templates than finetuned baselines: on Registration Form, LMDX$_{\text{PaLM 2-Small}}$ has a drop less than 5% Micro-F1 on Unseen Template compared to Single Template across data regimes, while baselines (LayoutLMv2) sees a drop between 19% and 27%.

On CORD (results in Table 3), we observe similar trends, highlighting the generalization of the results. At $|\mathcal{D}| = 10$, LMDX$_{\text{PaLM 2-Small}}$ is 4.03% from its peak performance attained at $|\mathcal{D}| = 800$, versus 22.34% for the strongest baseline LayoutLMv3$_{\text{LARGE}}$, showcasing LMDX's data efficiency.

**Performance on Hierarchical Entities.** As seen on Ad-Buy Form Mixed in Table 2, LMDX$_{\text{PaLM 2-Small}}$ has much higher Line Item F1 than the finetuned baselines for all data regimes. In particular, LMDX$_{\text{PaLM 2-Small}}$ has similar line item grouping performance at zero-shot than the best finetuned baseline at 200 train documents (21.21% versus 25.46% Line Item F1 respectively). With all the training data, LMDX$_{\text{PaLM 2-Small}}$ scores a 72.09% F1 on line item, an absolute improvement of 46.63% over the best baseline LayoutLMv2. Finally, LMDX$_{\text{PaLM 2-Small}}$, which encode spatial information, has much higher zero-shot Line Item F1 than large models baselines.

**Localization Accuracy** We compute the Localization Accuracy of LMDX$_{\text{PaLM 2-Small}}$ and all baselines that can localize entities using the formula: $Accuracy_{Localization} = \frac{N_{E+L}}{N_E}$ where $N_{E+L}$ is the number of entities correctly extracted and localized, and $N_E$ is the number of entities correctly extracted. This allows to evaluate the localization quality independently of the extraction quality. Since LMDX$_{\text{PaLM 2-Small}}$ localizes at the line level, localization verification is done at the line-level

as well, i.e. localization is considered correct if the prediction bounding box is covered by the groundtruth line-level bounding box by more than 80%. We present the results in the Localization Accuracy Columns in Table 2. Overall, LMDX$_{\text{PaLM 2-Small}}$ can localize its predictions reliably at the line-level with the segment identifiers, with 88%-93% accuracy at zero-shot, and 98%-99% in fine-tuned cases, which is slightly higher than LayoutLM/LayoutLMv2/LayoutLMv3/FormNet baselines that can localize their predictions.

Table 3: LMDX$_{\text{PaLM 2-Small}}$ results on CORD. Normalized Tree Edit Distance Accuracy is reported.

| Model | Localization | n-TED Accuracy | | | | | |
|---|---|---|---|---|---|---|---|
| | | $|\mathcal{D}| = 0$ | $|\mathcal{D}| = 10$ | $|\mathcal{D}| = 50$ | $|\mathcal{D}| = 100$ | $|\mathcal{D}| = 200$ | $|\mathcal{D}| = 800$ |
| LLaVA-v1.5-13B | ✗ | 4.78 | - | - | - | - | - |
| GPT-3.5 | ✗ | 58.25 | - | - | - | - | - |
| Donut | ✗ | 0.00 | 33.01 | 75.44 | 82.17 | 84.49 | 90.23 |
| LayoutLMv3$_{\text{LARGE}}$ | ✓ | 0.00 | 73.87 | 87.29 | 91.83 | 94.44 | 96.21 |
| **LMDX$_{\text{PaLM 2-Small}}$** | ✓ | **67.47** | **92.27** | **93.80** | **93.64** | **94.73** | **96.30** |

### 3.4 ABLATIONS

In this section, we ablate different facets of the LMDX methodology to highlight their relative importance. The results can be found in Table 4 and are discussed below. For all ablations, we evaluate on the VRDU Ad-Buy Form Mixed Template task, only changing the ablated facet.

Table 4: Ablations of Base Entity Extraction Training, Coordinate Tokens, and Sampling and their relative effects on extraction quality. All ablations are done on VRDU Ad-Buy Mixed Template.

| $|\mathcal{D}|$ | LMDX$_{\text{PaLM 2-Small}}$ | Without Base EE Training | | Without Coordinate Tokens | | Without Sampling Strategy | |
|---|---|---|---|---|---|---|---|
| | Micro-F1 | Micro-F1 | $\Delta$ (%) | Micro-F1 | $\Delta$ (%) | Micro-F1 | $\Delta$ (%) |
| **0** | 39.74 | 0.00 | -39.74 | 27.59 | -12.15 | 39.53 | -0.21 |
| **10** | 54.35 | 42.91 | -11.44 | 39.37 | -14.98 | 52.85 | -1.50 |
| **50** | 75.08 | 66.51 | -8.57 | 62.35 | -12.73 | 73.88 | -1.20 |
| **100** | 78.05 | 68.87 | -9.18 | 65.14 | -12.91 | 77.30 | -0.75 |
| **200** | 79.82 | 72.25 | -7.57 | 65.70 | -14.12 | 78.43 | -1.39 |

**Effects of Base Entity Extraction Training.** In this ablation, we remove the first stage training on the varied data mixture and directly finetune on the VRDU target task. As seen on columns 3-4 of Table 4, ablating that training stage leads to significant drop in extraction quality in finetuned scenarios and the complete loss of zero-shot extraction ability due to the model not respecting the extraction format, hence failing decoding. As the train set size increases, the degraded performance lessens from -11.44% to -7.57%, as the model learns the task and desired completion format.

**Effects of Coordinate Tokens.** In this ablation, we replace the coordinate tokens, which communicate the position of each line within the document, by the index of that line. This index still acts as a unique identifier for the line segment (required for entity localization and grounding) but does not communicate any position information. An example of a prompt with line index can be found in Appendix A.6. As can be seen on columns 5-6 of Table 4, the coordinate tokens are substantially important to the extraction quality, ranging from 12.15% to 14.98% absolute micro-F1 improvement across the data regimes.

**Effects of Sampling Strategy.** In this ablation, we discard our strategy of sampling $K = 16$ completions per chunk, and instead sample a single response. As seen in columns 7-8 of Table 4, this leads to a 0.21% to 1.5% drop in micro-F1. While overall minor for quality, the sampling strategy also allows to correct extraction format mistakes (parsing error rates are given in Appendix A.9), leading to a successful extraction on all documents within the benchmarks.

### 3.5 IN-CONTEXT LEARNING PERFORMANCE

In this section, we study how in-context learning (ICL) compares to finetuning for LMDX$_{\text{PaLM 2-Small}}$. To do so, we test two methodologies: *Random*, which randomly selects $|\mathcal{D}|$ documents and extrac-

tions from the train set, and *Nearest Neighbors*, which uses similarity based on SentenceT5 embeddings[2] (Ni et al., 2021) to retrieve $|\mathcal{D}|$ documents to add in the LLM context. The results on CORD are shown in Table 5, where n-TED is reported. Overall, while both methods increase the performance significantly, nearest neighbors has a clear advantage, matching the best random ICL performance with only a single in-context example (87.73% versus 87.37% n-TED), and matching the finetuned performance at $|\mathcal{D}| = 10$ examples (92.82% versus 92.27% n-TED), as examples from the same template are retrieved (see Appendix A.10 for example retrievals). Note that, beyond $|\mathcal{D}| = 10$ examples, the performance stops improving, as no more examples can fit in the context window of PaLM 2-Small.

Table 5: In-Context Learning results on CORD with different retrieval methods.

| ICL Method | $|\mathcal{D}| = 0$ | $|\mathcal{D}| = 1$ | $|\mathcal{D}| = 3$ | $|\mathcal{D}| = 5$ | $|\mathcal{D}| = 10$ | $|\mathcal{D}| = 20$ |
|---|---|---|---|---|---|---|
| Random | 67.47 | 74.96 | 84.88 | 86.47 | 87.26 | 87.37 |
| Nearest Neighbors | 67.47 | **87.73** | **90.98** | **92.28** | **92.82** | **92.75** |

### 3.6 ERROR ANALYSIS AND LIMITATIONS

In this section, we perform an error analysis on the test set to identify common error patterns of LMDX. A very common error type we observe is caused by OCR lines grouping multiple semantically different segments. We show two instance of those cases observed in LMDX$_{\text{PaLM 2-Small}}$ on the VRDU Ad-Buy Form in Figure 3. In the first example, prediction for the entity *line_item/program_desc* includes text from the previous column "Channel" along with the value in the column "Description". From the OCR line bounding boxes, we can see that these two columns are grouped as the same OCR line. In the second example, the model confuses between the adjacent keys "Invoice Period" and "Flight Dates" and extracts invoice dates as flight dates. Similar to the first example, OCR line bounding boxes show that the invoice dates and the key "Flight Dates" are grouped together in the same line although they are semantically different. As LMDX$_{\text{PaLM 2-Small}}$ uses only coarse line layout information ([$x_{\text{center}}$, $y_{\text{center}}$] with 100 quantization buckets), the model fails in these cases, which is a current limitation of LMDX. We believe that incorporating the image modality will help make LMDX more performant and robust to those OCR errors.

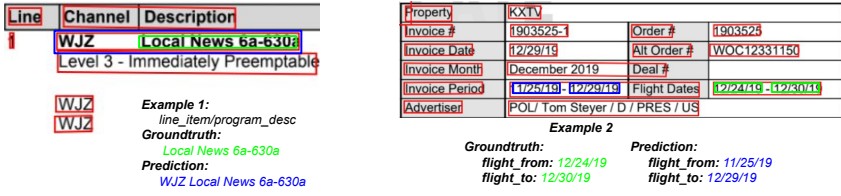

Figure 3: Typical error pattern of LMDX$_{\text{PaLM 2-Small}}$. In both examples, the detected OCR lines are shown in red, the model predicted entities are shown in blue, and the groundtruth entities are shown in green. In both cases, the detected OCR lines merge two semantically distinct segments, causing the model to wrongly associate them in its predictions.

## 4 CONCLUSION

In this paper, we have introduced LMDX, a methodology that enables using LLMs for information extraction on visually rich documents, setting a new state-of-the-art on public benchmarks VRDU and CORD. LMDX is the first methodology to allow the extraction of singular, repeated and hierarchical entities, while localizing the entities in the document. LMDX is data efficient, and even allows high quality extraction at zero-shot on entirely new document types and schemas. Nonetheless, since it relies on a LLM, LMDX is more resource-intensive than previous approaches, and its coordinate-as-tokens scheme requires long inputs and outputs. As future work, we will explore applying the methodology to open-source LLMs and adding the image modality to the system using Large Vision-Language Models.

---

[2]https://www.kaggle.com/models/google/sentence-t5/frameworks/tensorFlow2/variations/st5-base

## 5 REPRODUCIBILITY STATEMENT

In order to increase reproducibility, we've provided all details of the LMDX methodology. We've included our LLM prompts and completions in Appendix A.6, along with all our algorithms for chunking and decoding in Appendix A.1, A.2 and A.3. Furthermore, we've provided the exact target schemas used in our experiments in Appendix A.5. For CORD specifically, we've used a metric with a public implementation (`https://github.com/clovaai/donut/blob/master/donut/util.py`) and an easy to reproduce sampling strategy for the data-efficiency splits (first $D$ train documents). Finally, our baselines are publicly available (`https://github.com/microsoft/unilm/tree/master/layoutlmv3`, `https://github.com/clovaai/donut`) and thoroughly detailed in Appendix A.7 and A.8.

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

# A APPENDIX

## A.1 CHUNKING ALGORITHM

---
**Algorithm 1** Document Chunking
---
1: **function** CHUNK($D$, $L$, $F$)  ▷ $D$ is a document containing multiple pages. $L$ is token limit.
2:  ▷ $F$ is a function that outputs prompt token length given some segments (e.g. lines).
3:  $C = \phi$  ▷ $C$ is to record all produced chunks.
4:  **for** $i = 1$ to $|D.pages|$ **do**
5:  $S = D.pages[i].segments$
6:  **while** $S \neq \phi$ **do**
7:  **for** $j = |S|$ to 1 **do**  ▷ Start pruning from the end of the page.
8:  **if** $F(S[1:j]) \leq L$ **then**
9:  $C = C \cup \{S[1:j]\}$
10:  $S = S[j+1:|S|]$  ▷ Continue to work on the rest of the segments.
11:  Exit for loop
12:  **end if**
13:  **end for**
14:  **end while**
15:  **end for**
16:  **return** $C$
17: **end function**
---

## A.2 ENTITY VALUE PARSING ALGORITHM

---
**Algorithm 2** Entity Value Parsing
---
1: **function** PARSEENTITYVALUE($D$, $E$)  ▷ $D$ is a document chunk.
2:  ▷ $E$ is raw extraction results for one entity type parsed from one LLM sample.
3:  $G = \phi$  ▷ $G$ is to record all parsed entity values.
4:  $R = Regex(``(\backslash d\backslash d\backslash|\backslash d\backslash d)")$  ▷ $R$ is a regex that captures the segment identifiers.
5:  $M = \{``s.x|s.y" \mapsto s|s \in D.segments\}$  ▷ $M$ holds a mapping between segment id and segment.
6:  **for** $i = 1$ to $|E|$ **do**
7:  $W = \phi$  ▷ $W$ is to hold all words for this entity.
8:  $P = R.split(E[i])$  ▷ P is expected to be interleaved text values and segment ids.
9:  **for** $j = 1$ to $|P|/2$ **do**
10:  **if** $P[j*2] \notin M$ **then**
11:  Go to next $i$  ▷ Segment ID is hallucinated. Grounding failure.
12:  **end if**
13:  $S = M[P[j*2]]$  ▷ Retrieve the stored segment from $M$ with parsed segment ID.
14:  $T = P[j*2-1]$  ▷ $T$ is to hold the predicted text.
15:  **if** $T$ not substring of $S$ **then**
16:  Go to next $i$  ▷ Grounding failure, skip the current entity.
17:  **end if**
18:  $W = W \cup (S \cap T)$
19:  **end for**
20:  $G'.value = \bigcup_{w \in W} w.text\_value$  ▷ $G'$ is to hold the entity to return.
21:  $G'.bounding\_box = \{\min(b.x), \min(b.y), \max(b.x), \max(b.y)\}_{w \in W, b=w.bounding\_box}$
22:  $G = G \cup \{G'\}$
23:  **end for**
24:  **return** $G$
25: **end function**
---

## A.3 Decoding Algorithm

---

**Algorithm 3** Responses Decoding

---

1: **function** DECODEFORTYPE($J$, $T$, $D$)                                                    $\triangleright$ $J$ is one or more JSON objects.
2:                                                                                                $\triangleright$ $T$ is an entity type.
3:                                                                                                $\triangleright$ $D$ is a document chunk.
4:     $E = \phi$                                                                    $\triangleright$ $E$ is to record all parsed and grounded entities.
5:     **for** $j = 1$ to $|J|$ **do**
6:         $J' = J[j][T.type]$                                    $\triangleright$ $J'$ is to hold entities for T's type before grounding.
7:         **if** $T.subtypes = \phi$ **then**                                            $\triangleright$ $T$ is leaf entity type.
8:             $E = E \cup ParseEntityValue(D, J')$
9:         **else**                                                            $\triangleright$ $T$ is hierarchical entity type.
10:            $E'.subtypes = \bigcup_{T' \in T.subtypes} DecodeForType(J', T', D)$        $\triangleright$ $E'$ is hierarchical entity.
11:            $E = E \cup \{E'\}$
12:        **end if**
13:    **end for**
14:    **return** $E$
15: **end function**
16:
17: **function** MAJORITYVOTING($T$, $E$)                                        $\triangleright$ $T$ is an entity type.
18:                                                        $\triangleright$ $E$ is a 2D vector of entities of type $T$ from all LLM responses.
19:    $V = [0, 0, ..., 0] \in \mathbb{R}^{|E|}$                                            $\triangleright$ $V$ is to record all votes.
20:    $L = \{T\}$
21:    **while** $L \neq \phi$ **do**
22:        $T' = L[0]$
23:        $E' = \phi$
24:        **for** $j = 1$ to $|E|$ **do**
25:            $E' = E' \cup \{e | e \in E[j], e.type = T'\}$            $\triangleright$ $E'[j]$ holds entities with type $T'$ from E[j].
26:        **end for**
27:        **for** $i = 1$ to $|E'|$ - 1 **do**
28:            **for** $j = i + 1$ to $|E'|$ **do**
29:                **if** $E'[i] = E'[j]$ **then**
30:                    $V[i] = V[i] + 1$
31:                    $V[j] = V[j] + 1$
32:                **end if**
33:            **end for**
34:        **end for**
35:        $L = L[1 : |L|]$                                        $\triangleright$ Remove $T'$ and inject its sub-types for recursion.
36:        $L = L \cup T'.subtypes$
37:    **end while**
38:    **return** $E[argmax(V)]$                                    $\triangleright$ Return the entity values with the highest votes.
39: **end function**
40:
41: **function** DECODEALLSAMPLES($S$, $T$, $D$)                        $\triangleright$ $S$ is all LLM response samples on $D$.
42:                                                                        $\triangleright$ $T$ is a list of entity types.
43:                                                                        $\triangleright$ $D$ is a document chunk.
44:    **return** $\bigcup_{T' \in T} MajorityVoting(\bigcup_{S' \in S} DecodeForType(ParseJson(S'), T', D))$
45: **end function**

---

## A.4 TOKEN LENGTH STATISTICS

Table 6 details the token length (50th and 99th percentiles) of the prompt and completion targets for the train split of datasets used in our experiments. We select the line level segment, 2 coordinate scheme, no JSON indentation so that all datasets fit within our 6144 prompt token length and 2048 output token length.

Table 6: Prompt and target token length of different coordinate-as-tokens schemes on VRDU and CORD benchmarks, using the vocabulary of PaLM 2-S. We vary the number of coordinates and their quantization buckets in the localization tags, the segment level (e.g. line versus word), chunking style (e.g. page versus max input tokens) and JSON indentation in the schema and completion targets.

**VRDU Ad-Buy Form**

| # Coord. | # Quant. | Segment | Chunking | JSON Indent | Input 50th | Input 99th | Target 50th | Target 99th |
|---|---|---|---|---|---|---|---|---|
| 2 | 100 | Line | Page | None | 2377 | 3920 | 602 | 1916 |
| 2 | 100 | Word | Page | None | 3865 | 13978 | 718 | 2328 |
| 4 | 100 | Line | Page | None | 3329 | 5284 | 777 | 2473 |
| 2 | 1000 | Line | Page | None | 2687 | 4322 | 660 | 2095 |
| 2 | 100 | Line | Page | 4 | 2417 | 3328 | 689 | 2234 |
| 2 | 100 | Line | 6144 tokens | None | 2377 | 3920 | 602 | 1916 |

**VRDU Registration Form**

| # Coord. | # Quant. | Segment | Chunking | JSON Indent | Input 50th | Input 99th | Target 50th | Target 99th |
|---|---|---|---|---|---|---|---|---|
| 2 | 100 | Line | Page | None | 963 | 1578 | 79 | 147 |
| 2 | 100 | Word | Page | None | 3083 | 5196 | 101 | 349 |
| 4 | 100 | Line | Page | None | 1232 | 2017 | 91 | 177 |
| 2 | 1000 | Line | Page | None | 1052 | 1723 | 83 | 155 |
| 2 | 100 | Line | Page | 4 | 977 | 1592 | 92 | 160 |
| 2 | 100 | Line | 6144 tokens | None | 963 | 1578 | 79 | 147 |

**CORD**

| # Coord. | # Quant. | Segment | Chunking | JSON Indent | Input 50th | Input 99th | Target 50th | Target 99th |
|---|---|---|---|---|---|---|---|---|
| 2 | 100 | Line | Page | None | 342 | 869 | 355 | 1495 |
| 2 | 100 | Word | Page | None | 396 | 1067 | 375 | 1638 |
| 4 | 100 | Line | Page | None | 408 | 1139 | 422 | 1801 |
| 2 | 1000 | Line | Page | None | 364 | 959 | 376 | 1957 |
| 2 | 100 | Line | Page | 4 | 411 | 938 | 474 | 1997 |
| 2 | 100 | Line | 6144 tokens | None | 342 | 869 | 355 | 1495 |

## A.5 SCHEMAS

In this section, we present the schemas used for the experiments of this paper. The schema for VRDU Ad-Buy Form, VRDU Registration Form, and CORD can be found in Figure 4, Figure 5 and Figure 6 respectively.

```
{
  "advertiser": "",
  "agency": "",
  "contract_num": "",
  "flight_from": "",
  "flight_to": "",
  "gross_amount": "",
  "line_item": [
    {
      "channel": "",
      "program_desc": "",
      "program_end_date": "",
      "program_start_date": "",
      "sub_amount": ""
    }
  ],
  "product": "",
  "property": "",
  "tv_address": ""
}
```

Figure 4: VRDU Ad-Buy Form Schema.

```
{
  "file_date": "",
  "foreign_principle_name": "",
  "registrant_name": "",
  "registration_num": "",
  "signer_name": "",
  "signer_title": ""
}
```

Figure 5: VRDU Registration Form Schema.

```
{
  "line_item": [ # menu
    {
      "discount_price": "", # menu.discountprice
      "identifier": "", # menu.num
      "name": "", # menu.nm
      "other": "", # menu.etc
      "quantity": "", # menu.qty
      "sub_name": [], # menu.sub_nm
      "sub_price": [], # menu.sub_price
      "sub_quantity": [], # menu.sub_qty
      "subtotal_price": "", # menu.itemsubtotal
      "total_price": "", # menu.price
      "unit_price": "" # menu.unitprice
    }
  ],
  "subtotal": { # subtotal
    "discount_price": "", # subtotal.discount_price
    "other": [], # subtotal.etc
    "service_price": "", # subtotal.service_price
    "subtotal_price": [], # subtotal.subtotal_price
    "tax_price": [] # subtotal.tax_price
  },
  "total": { # total
    "cash_price": [], # total.cashprice
    "change_price": "", # total.changeprice
    "credit_card_price": "", # total.creditcardprice
    "emoney_price": "", # total.emoneyprice
    "line_item_quantity_count": "", # total.menuqty_cnt
    "line_item_type_count": "", # total.menutype_cnt
    "other": "", # total.total_etc
    "total_price": [] # total.total_price
  }
}
```

Figure 6: CORD Schema. Note that the original entity types (shown as comments) have been renamed to more semantically meaningful names.

A.6  SAMPLE PROMPTS AND COMPLETIONS

In this section, we present example of LMDX prompts and completions from the LLM on the VRDU Ad-Buy dataset to better showcase the format used. Figure 7 shows the original document with the line bounding boxes from OCR, Figure 8 shows the corresponding prompt and completion on that document with coordinate segment identifiers, and Figure 9 shows the same prompt and completion, but with line index segment identifiers (used in ablation studies to showcase how the LLM can interpret the layout).

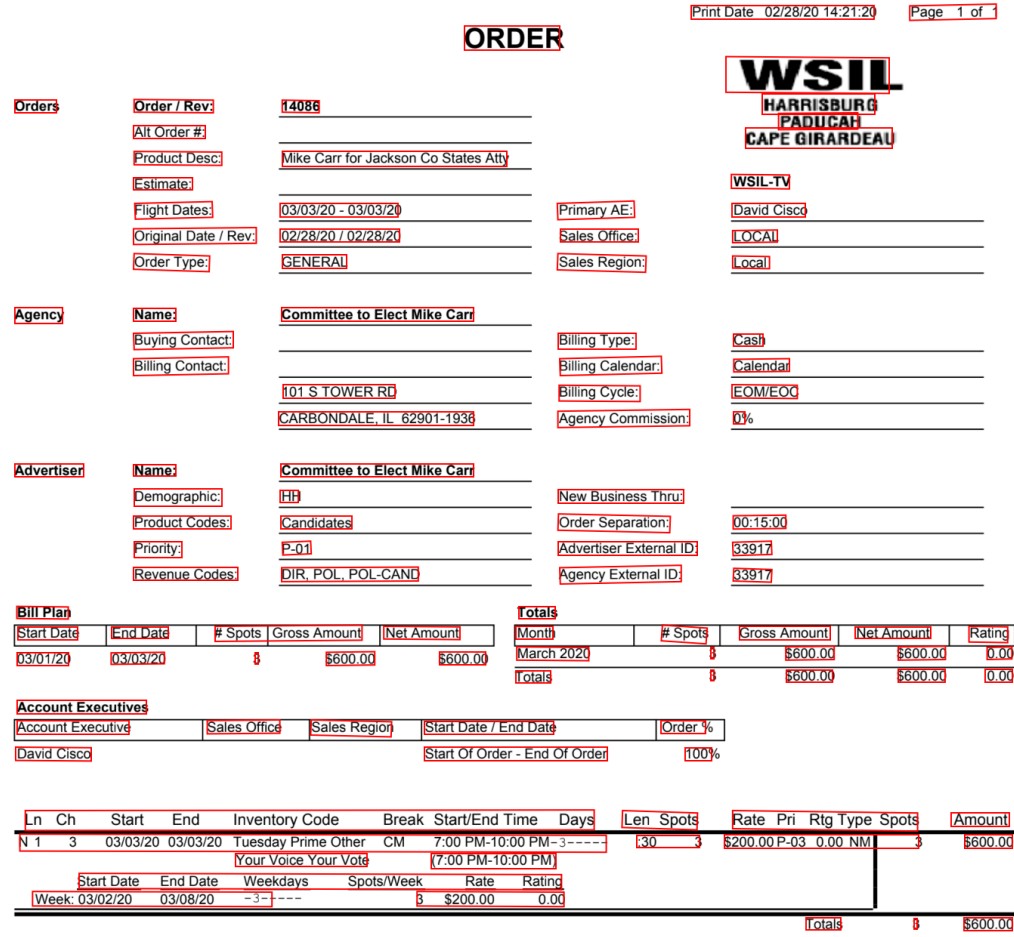

Figure 7: Example document from VRDU Ad-Buy Form. The line bounding boxes detected by OCR are shown in red.

```
============================== Prompt ==============================

<Document>
Print Date 02/28/20 14:21:20 75|02
Page 1 of 1 91|02
ORDER 49|04
WSIL 77|07
Orders 04|09
Order/Rev: 17|09
14086 29|09
HARRISBURG 78|09
PADUCAH 78|10
CAPE GIRARDEAU 78|12
Alt Order #: 17|11
Product Desc: 18|13
Mike Carr for Jackson Co States Atty 38|13
Estimate: 16|15
WSIL-TV 73|15
Flight Dates: 17|17
03/03/20 - 03/03/20 33|17
Primary AE: 57|17
David Cisco 73|17
Original Date/ Rev: 19|19
02/28/20/02/28/20 33|19
...
Start Date End Date Weekdays Spots/Week Rate Rating 31|66
Week: 03/02/20 03/08/20 -3-- 15|67
3 $200.00 0.00 47|67
Totals 79|69
3 87|69
$600.00 94|69
</Document>
<Task>
From the document, extract the text values and tags of the following
entities:
{"advertiser": "", "agency": "", "contract_num": "", "flight_from": "",
"flight_to": "", "gross_amount": "", "line_item": [{"channel": "",
"program_desc": "", "program_end_date": "", "program_start_date": "",
"sub_amount": ""}], "product": "", "property": "", "tv_address": ""}
</Task>
<Extraction>

============================== Completion ==============================

{"advertiser": "Committee to Elect Mike Carr 37|36", "agency":
"Committee to Elect Mike Carr 37|24", "contract_num": "14086 29|09",
"flight_from": "03/03/20 33|17", "flight_to": "03/03/20 33|17",
"gross_amount": "$600.00 77|51", "line_item": [{"channel": "3 31|63",
"program_desc": "Tuesday Prime 31|63\nYour Voice Your Vote 29|64",
"program_end_date": "03/03/20 31|63", "program_start_date":
"03/03/20 31|63", "sub_amount": "$600.00 94|63"}], "product":
"Mike Carr for Jackson Co States Atty 38|13", "property": "WSIL 77|07",
"tv_address": null}
</Extraction>
```

Figure 8: VRDU Ad-Buy Form sample prompt and completion with 2 coordinates for segment identifier. The document is truncated for easier visualization.

```
============================== Prompt ==============================

<Document>
Print Date 02/28/20 14:21:20 <0>
Page 1 of 1 <1>
ORDER <2>
WSIL <3>
Orders <4>
Order/Rev: <5>
14086 <6>
HARRISBURG <7>
PADUCAH <8>
CAPE GIRARDEAU <9>
Alt Order #: <10>
Product Desc: <11>
Mike Carr for Jackson Co States Atty <12>
Estimate: <13>
WSIL-TV <14>
Flight Dates: <15>
03/03/20 - 03/03/20 <16>
Primary AE: <17>
David Cisco <18>
Original Date/ Rev: <19>
02/28/20/02/28/20 <20>
...
Start Date End Date Weekdays Spots/Week Rate Rating <105>
Week: 03/02/20 03/08/20 -3-- <106>
3 $200.00 0.00 <107>
Totals <108>
3 <109>
$600.00 <110>
</Document>
<Task>
From the document, extract the text values and tags of the following
entities:
{"advertiser": "", "agency": "", "contract_num": "", "flight_from": "",
"flight_to": "", "gross_amount": "", "line_item": [{"channel": "",
"program_desc": "", "program_end_date": "", "program_start_date": "",
"sub_amount": ""}], "product": "", "property": "", "tv_address": ""}
</Task>
<Extraction>

============================== Completion ==============================

{"advertiser": "Committee to Elect Mike Carr <44>", "agency":
"Committee to Elect Mike Carr <29>", "contract_num": "14086 <6>",
"flight_from": "03/03/20 <16>", "flight_to": "03/03/20 <16>",
"gross_amount": "$600.00 <84>", "line_item": [{"channel": "3 <102>",
"program_desc": "Tuesday Prime Other CM <102>", "program_end_date":
"03/03/20 <102>", "program_start_date": "03/03/20 <102>",
"sub_amount": "$600.00 <100>"}], "product": "Mike Carr for Jackson Co
States Atty <12>", "property": "WSIL-TV <14>", "tv_address": null}
</Extraction>
```

Figure 9: VRDU Ad-Buy Form sample prompt and completion with line index for segment identifier, which does not communicate layout information. The document is truncated for easier visualization.

## A.7 COMMON BASELINES DETAILS

We compare LMDX to other Large Model baselines on all benchmarks in the zero-shot context. Those baselines are detailed below.

**GPT-3.5 Baseline.**    We evaluate the zero-shot extraction ability of GPT-3.5, a strong LLM baseline. To do so, we prompt it with the raw OCR text (no coordinate tokens or segment identifier like for LMDX), and extraction instructions alongside the schema in JSON format. We then parse the completions as JSON to get the predicted entities directly. Note that GPT-3.5's predicted entities are also not localized within the document. A sample prompt can be observed in Figure 10.

**LLaVA-v1.5-13B Baseline.**    We evaluate the zero-shot extraction ability of LLaVA-v1.5-13B, a strong vision-text large model. The prompt includes task description, instructions and target schema represented in JSON format as text input and the document page as image input. We provide examples of valid JSON values in the task instructions. For each page of a test document, we infer the extraction in a JSON format. We merge the individual page JSONs to obtain the final extraction for a document. Overall, in the the zero-shot setting, we notice the JSON parse error rate of the LLM completions is 10% which is higher than that of LMDX (as seen in the Appendix A.9). Along with invalid JSON format, the model errors also include several OCR errors and hallucinations of entity values. Note that LLaVA-v1.5-13B's predicted entities are also not localized within the document. A sample prompt can be observed in Figure 11.

```
${RAW_OCR_TEXT}

Given the document, extract the text value of the entities included in
the schema in json format.
- The extraction must respect the JSON schema.
- Only extract entities specified in the schema. Do not skip any
entity types.
- The values must only include text found in the document.
- Use null or [] for missing entity types.
- Do not indent the json you produce.
- Examples of valid string value format: "$ 1234.50", "John Do", null.
- Examples of valid list value format: ["$ 1234.50", "John Do"], [].

Schema: {"file_date": "", "foreign_principle_name": "",
"registrant_name": "", "registration_num": "", "signer_name": "",
"signer_title": ""}
```json
```

Figure 10: Sample prompt for GPT-3.5 baseline for VRDU Registration Form.

```
${DOCUMENT_IMAGE}

Given the document, extract the text value of the entities included in
the schema in json format.
- The extraction must respect the JSON schema.
- Only extract entities specified in the schema. Do not skip any

entity types.
- The values must only include text found in the document.
- Use null or [] for missing entity types.
- Do not indent the json you produce.
- Examples of valid string value format: "$ 1234.50", "John Do", null.
- Examples of valid list value format: ["$ 1234.50", "John Do"], [].

Schema: {"file_date": "", "foreign_principle_name": "",
"registrant_name": "", "registration_num": "", "signer_name": "",
"signer_title": ""}
```json
```

Figure 11: Sample prompt for LLaVA-v1.5-13B baseline for VRDU Registration Form.

## A.8    CORD BASELINES DETAILS

**LayoutLMv3$_{\text{LARGE}}$    Baseline.**    We    follow    the    released    implementation[3]    for    the LayoutLMv3$_{\text{LARGE}}$    model    and    the    training    protocol    described    in    Huang    et    al.    (2022)    as closely as possible. In particular, we train the model for $80$ epochs for each experiment on CORD (namely, $10, 50, 100, 200$, and $800$-document training sets), on the IOB tags of the leaf entities. One difference in our training is that, due to computational resource constraints, we use $batch\_size = 8$ and $learning\_rate = 2 \cdot 10^{-5}$.

As the LayoutLMv3 model can only extract leaf entities, we design and heavily optimize a heuristic algorithm to group the leaf entities into hierarchical entities *menu*, *subtotal* and *total*. The best heuristics we could find are as follows:

- For the *subtotal* and *total* hierarchical entity types, since they appear only once per document, we group all their extracted sub-entities under a single *subtotal* and *total* entity, respectively.

- For *menu* hierarchical entity type, we observe that those entities usually occur multiple times on a document, and each *menu* has at most one *nm*, *num*, *unitprice*, *cnt*, *discountprice*, *price*, *itemsubtotal*, *etc* sub-entities and potentially multiple *sub_nm*, *sub_price* and *sub_cnt* sub-entities. We also notice that the sub-entities aligned horizontally overwhelmingly belong to the same *menu* entity, and a *menu* entity can sometimes span over two or more consecutive horizontal lines. To leverage those observations, we perform a two-step grouping process for *menu* entities. First, we merge the extracted leaf sub-entities into horizontal groups, where a threshold of $0.5$ on the intersection-over-union of the Y-axis was used for the determination of horizontal alignment. Second, we further merge the *consecutive* horizontal groups into *menu* entities, if and only if the horizontal groups do not have type duplication in any of the *nm*, *num*, *unitprice*, *cnt*, *discountprice*, *price*, *itemsubtotal*, and *etc* sub-entities (namely, those sub-entities only show up in at most one of the consecutive horizontal groups to be merged). We allow duplication of *sub_nm*, *sub_price* and *sub_cnt* sub-entity types. After those two steps, we obtain the final *menu* entities.

**Donut Baseline.**    We follow Donut released implementation[4] for the Donut benchmarking results on CORD. We use the default training configuration for all experiments on CORD (namely, $10$, $50, 100, 200$, and $800$-document training sets), with the following difference: we reduce batch size from 8 to 4 due to computational resource constraints, and increase the number of train epochs from 30 to 60. For each experiment, checkpoint with the lowest loss on the dev set is selected and we report performance on test set. Normalized Tree Edit Distance accuracy scores produced by Donut evaluation code are reported (similar to all our other models).

---

[3]https://github.com/microsoft/unilm/tree/master/layoutlmv3
[4]https://github.com/clovaai/donut

## A.9 COMPLETION PARSING ERROR RATES

In this section, we report the various completion parsing error types and their occurrence rates.

**Invalid JSON Formatting.** This error refers to cases for which Python's `json.loads(completion)` fails on a LLM's completion. As observed in Table 7, the JSON parsing error rate is below 0.3% in all training settings.

**Invalid Entity Value Format.** This error refers to cases where the leaf entity value does not follow the expected *"<text-segment-1> XX|YY <text-segment-2> XX|YY"* format. As observed in Table 7, the Invalid Entity Value Format Rate is below 0.05% in all training settings.

**Hallucination / Entity Text Not Found.** This error refers to cases where the segment identifier is valid, but the entity text does not appear on the predicted segment (hallucination). As observed in Table 7, the Entity Text Not Found error rate is below 0.6% in all training settings. As part of LMDX methodology, we discard any prediction whose text does not appear on the specified segment, ensuring we discard all hallucination.

Note that those numbers are computed at the completion level. Since multiple completions are sampled for each document chunk, the sampling scheme allows for correcting those errors and no document in the benchmarks fail extraction.

Table 7: Breakdown of parsing error rates from LMDX$_{\text{PaLM 2-Small}}$ responses on VRDU Ad-Buy Mixed and CORD datasets.

| $|\mathcal{D}|$ | Dataset | Invalid JSON | Invalid Entity Value Format | Entity Text Not Found |
|---|---|---|---|---|
| **0** | Ad-buy Form | 0.18% | 0.04% | 0.59% |
| | CORD | 0.00% | 0.00% | 0.00% |
| **10** | Ad-buy Form | 0.27% | 0.04% | 0.44% |
| | CORD | 0.00% | 0.00% | 0.00% |
| **50** | Ad-buy Form | 0.24% | 0.00% | 0.17% |
| | CORD | 0.06% | 0.00% | 0.00% |
| **100** | Ad-buy Form | 0.24% | 0.00% | 0.13% |
| | CORD | 0.00% | 0.03% | 0.00% |
| **200** | Ad-buy Form | 0.25% | 0.00% | 0.09% |
| | CORD | 0.00% | 0.00% | 0.00% |

## A.10   IN-CONTEXT LEARNING WITH NEAREST NEIGHBORS

In our study, nearest neighbors leads to a significant quality gain over randomly selecting exemplars. In this section, we explore why that is the case in the context of VRD information extraction. Figures 12, 13 and 14 show typical retrievals using sentenceT5 embeddings on the OCR text for similarity. Unsurprisingly, nearest neighbors works well as it retrieves examplars from the same template as the target document, i.e. from the same merchant in the case of CORD documents (store and restaurant receipts). As those examples share the same layout, same boilerplate text, and same entities, it makes it a lot easier for the model to understand the correct extraction pattern.

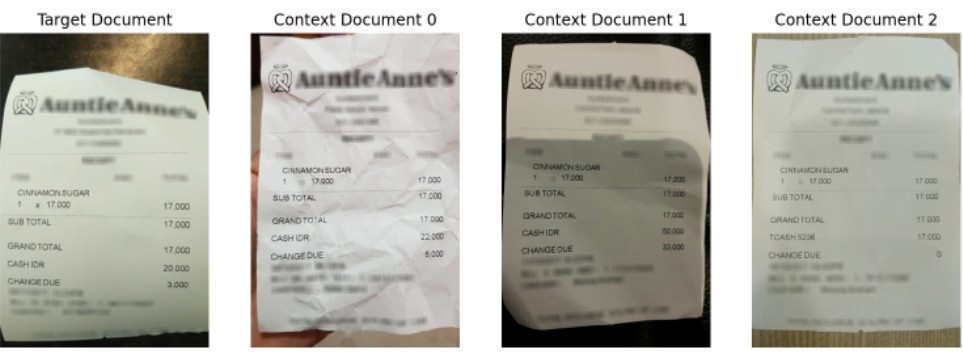

Figure 12: Nearest Neighbors on CORD, Example 1, retrieving examplars from the same merchant.

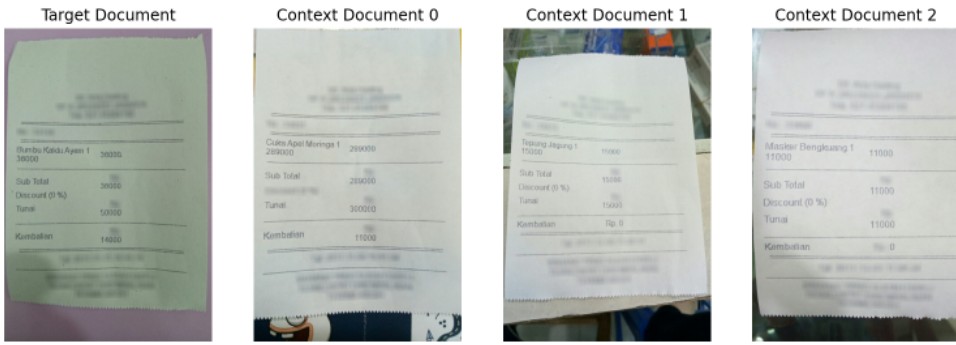

Figure 13: Nearest Neighbors on CORD, Example 2, retrieving examplars from the same merchant.

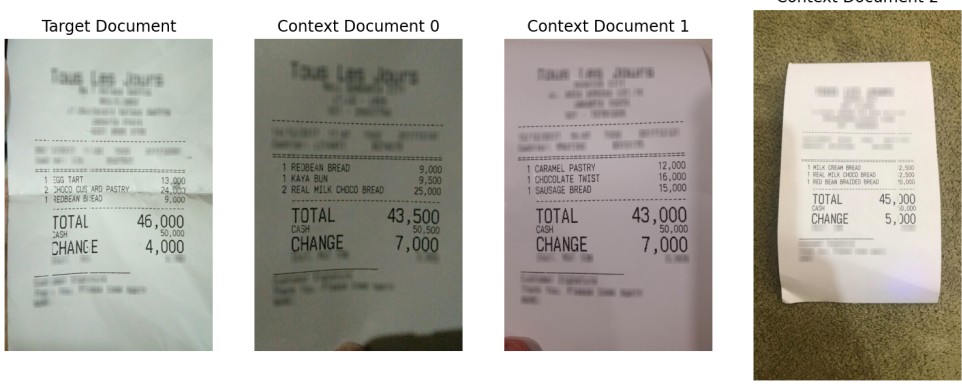

Figure 14: Nearest Neighbors on CORD, Example 3, retrieving examplars from the same merchant.

