# OpenReview forum: "LMDX: Language Model-based Document Information Extraction and Localization"
_ICLR.cc/2024/Conference — ICLR 2024 Conference Withdrawn Submission_

### Official Review · Reviewer_5Y8u · 2023-10-26

**Soundness:** 2 fair
**Presentation:** 3 good
**Contribution:** 3 good
**Rating:** 5
**Confidence:** 3

**Summary:**

This paper proposes a method to extract information from visually rich documents with the information and their position. The method extracts information from the OCR'ed document by prompting the PaLM2-S LLM to perform completion tasks. With the LLM with entity extraction training, the model can achieve strong performance even with no training data, comparable to or better than a few baselines. With a few-shot setting, the method shows a high performance with a large margin compared to existing methods.

**Strengths:**

- This paper proposes a novel method to document information extraction from visually rich documents using LLMs.
- The absolute performance is greatly higher than existing methods, and some of the proposed enhancements are shown effective through the ablation study.
- The paper is well-written with the details of algorithms, schemas, sample outputs, etc.

**Weaknesses:**

- The model's performance highly depends on the off-the-shelf OCR and PaLM2-S large language model, but they are unavailable, so the results are not reproducible. Also, there is no detailed explanation or evaluation of these modules.
- The authors mention the support of the hierarchical entity and entity localization, but their effect is not directly evaluated since there is no evaluation without them.

**Questions:**

- Is there any performance assessment of the OCR model?
- How does the model differ from baselines in, e.g., the number of parameters and runtime? Although the authors remark on using open-source LLMs as future work, how is it difficult to run the model with publicly accessible OCR and existing LLMs?
- Did the authors try other prompts, schema, or target formats during the development? How are the current settings chosen?

**Details Of Ethics Concerns:**

No problem.

---

> ### Author Response · Authors · 2023-11-20
> **Response to reviewer 5Y8U (Part 1)**
>
> We would like to thank reviewer 5Y8u for taking the time to read our paper in detail and giving valuable feedback! Please see our response below:
>
> > The model's performance highly depends on the off-the-shelf OCR and PaLM2-S large language model, but they are unavailable, so the results are not reproducible. Also, there is no detailed explanation or evaluation of these modules.
>
> **[Response]** We agree we did not make this clear in the submitted paper, but all our experiments use the OCR provided by the benchmark for all models and baselines. As such the reported performance improvements are entirely due to the model and LMDX methodology, and we can fairly compare it to the baselines. We will make this clearer in the next version of the paper.
>
> > The authors mention the support of the hierarchical entity and entity localization, but their effect is not directly evaluated since there is no evaluation without them.
>
> **[Response]** With regards to hierarchical entity evaluation, we believe this is directly evaluated: we highlight the F1 score on line_item entity (the hierarchical entity in VRDU) for all models and baselines in Table 2 and a provide a discussion for it in Section 3.3 *Performance on Hierarchical Entities*. The table below synthetizes this information and shows the line_item F1:
>
> | Dataset Size | 0     | 10    | 50    | 100   | 200   |
> |--------------|-------|-------|-------|-------|-------|
> | FormNet      | N/A   | 5.72  | 19.06 | 18.80 | 21.86 |
> | LayoutLM     | N/A   | 6.95  | 19.50 | 21.26 | 23.90 |
> | LayoutLMv2   | N/A   | 9.96  | 20.98 | 23.52 | 25.46 |
> | LayoutLMv3   | N/A   | 5.92  | 19.53 | 22.08 | 24.51 |
> | LMDX         | 21.21 | 39.35 | 65.42 | 69.77 | 72.09 |
>
> Overall, previous SOTA methods (LayoutLM / FormNet) plateau around ~20-25% F1 due to their reliance on heuristics that do not benefit from additional training data, unlike LMDX. Please let us know if there are more that would be useful to show with regards to hierarchical entity evaluation.
>
> We also acknowledge that in the submitted paper version, entity localization is not directly/independently evaluated, and we thank reviewer 5Y8U for pointing it out. For all models and baselines that can do entity localization, on VRDU Registration Form and Ad-Buy Form Mixed, we've computed the localization accuracy, following the formula below:
>
> $$Accuracy_{Localization} = TotalEntityCount_{CorrectlyExtractedAndLocalized} \div TotalEntityCount_{CorrectlyExtracted} $$
>
> Since LMDX localizes at the line level, localization verification is done at the line-level as well, i.e. localization is considered correct if the prediction bounding box is covered by the groundtruth line-level bounding box by more than 80%. See the results below.
>
> VRDU Registration Form Mixed:
>
> | Train Dataset Size / Model | LayoutLM | LayoutLMv2 | LayoutLMv3 | **LMDX (Ours)**   |
> |----------------------|----------|------------|------------|--------|
> | 0 doc                | N/A      | N/A        | N/A        | 93.21% |
> | 10 doc               |   98.71% |     99.00% |     99.20% | 99.75% |
> | 50 doc               |   99.69% |     99.54% |     99.39% | 99.87% |
> | 100 doc              |   99.63% |     99.72% |     99.72% | 99.92% |
> | 200 doc              |   99.69% |     99.75% |     99.67% | 99.87% |
>
> VRDU Ad-Buy Form Mixed:
>
> | Train Dataset Size / Model | LayoutLM | LayoutLMv2 | LayoutLMv3 | **LMDX (Ours)**   |
> |----------------------|----------|------------|------------|--------|
> | 0 doc                | N/A      | N/A        | N/A        | 88.18% |
> | 10 doc               |   92.60% |     93.95% |     90.68% | 94.51% |
> | 50 doc               |   95.24% |     95.64% |     95.28% | 98.28% |
> | 100 doc              |   95.09% |     95.72% |     95.88% | 98.69% |
> | 200 doc              |   95.38% |     95.78% |     95.95% | 98.65% |
>
> This evaluates the localization quality independently of the extraction quality. We will update the next version of the paper to include those results.
>
> > Is there any performance assessment of the OCR model?
>
> **[Response]** In all our experiments, we simply use the publicly provided benchmark's OCR for all models and baselines, thus we do not explicitly evaluate the performance of the OCR model. This means that the quality improvements showcased come from the parsing model (powered by LMDX, FormNet, LayoutLMv1/2/3, etc) and not the OCR itself, leading to a fair comparison between LMDX and baselines.
>
> For details regarding the OCR, the VRDU authors mention they used Google Vision OCR (per page 5 of [1]).
>
> [1] Wang, Zilong, Yichao Zhou, Wei Wei, Chen-Yu Lee, and Sandeep Tata. "Vrdu: A benchmark for visually-rich document understanding." In Proceedings of the 29th ACM SIGKDD Conference on Knowledge Discovery and Data Mining, pp. 5184-5193. 2023.

---

> > ### Author Response · Authors · 2023-11-20
> > **Response to reviewer 5Y8U (Part 2)**
> >
> > > Is there any performance assessment of the OCR model?
> >
> > **[Response]** In all our experiments, we simply use the publicly provided benchmark's OCR for all models and baselines, thus we do not explicitly evaluate the performance of the OCR model. This means that the quality improvements showcased come from the parsing model (powered by LMDX, FormNet, LayoutLMv1/2/3, etc) and not the OCR itself, leading to a fair comparison between LMDX and baselines.
> >
> > For details regarding the OCR, the VRDU authors mention they used Google Vision OCR (per page 5 of [1]).
> >
> > [1] Wang, Zilong, Yichao Zhou, Wei Wei, Chen-Yu Lee, and Sandeep Tata. "Vrdu: A benchmark for visually-rich document understanding." In Proceedings of the 29th ACM SIGKDD Conference on Knowledge Discovery and Data Mining, pp. 5184-5193. 2023.
> >
> > > How does the model differ from baselines in, e.g., the number of parameters and runtime?
> >
> > **[Response]** Unfortunately, our organization does not allow us to disclose certain details of PaLM 2 like its parameters or runtime, so this was not an analysis we could provide. We however would like to emphasize that we use the small version of PaLM 2 (PaLM 2-S), and that only a single inference call is needed per document chunk (most documents are single chunk, and chunks are processed in parallel).
> >
> > > Although the authors remark on using open-source LLMs as future work, how is it difficult to run the model with publicly accessible OCR and existing LLMs?
> >
> > **[Response]** Regarding using open-source LLMs, it is not particularly technically difficult as long as one has access to enough compute resources for SFT, and an open source model that has long enough input/output sequence length support. For purpose of the paper, running the experiments on the current set of benchmarks is expensive, and requires 81 fine-tuning runs to get all the numbers (hence why we focused on a single LLM, PaLM-2 S, in the manuscript).
> >
> > Regarding OCR, we are already using the publicly available OCRs from the benchmarks.
> >
> > > Did the authors try other prompts, schema, or target formats during the development? How are the current settings chosen?
> >
> > **[Response]** Yes! We did try multiple prompt formats during development. In particular, we tried:
> >
> > * 2 coordinates vs 4 coordinates or each segment: using 4 coordinates led to a very small quality decrease compared to 2 coordinates.
> > * line level vs word level segment: word-level segment performed worse than line-level segment. We posit that this is due to the fact that word-level segments change the text sequence so much, it starts being very different from what the LLM was initially pretrained on (most tokens start being coordinate tokens), and as such the LLM struggles more to interpret the sequence.
> > * 100 coordinate quantization buckets vs 1000 quantization buckets: this was overall neutral quality-wise but 1000 used more tokens overall in its prompt and completions.
> > * Indent the json for the schema and completion : this was quality-neutral. Hence we chose no indentation to save on prompt/completion tokens (a 14% save, leading to faster and cheaper inference).
> > * More detailed instruction in the prompt (the {TASK_DESCRIPTION} in Figure 2). This was quality-neutral after the base entity extraction training (Figure 3).
> > * Trying document first versus last in the prompt: having the document before was a slight quality gain.
> >
> > Thus in the end, since it worked best with PaLM 2-Small, we went with 2 coordinates, line-level segment, 100 coordinate quantization buckets and no json indentation, with which we ran the experiments on CORD and VRDU benchmarks.
> >
> > Let us know if that addresses your questions/concerns, and thank you again for reviewing our paper.

---

> > > ### Comment · Reviewer_5Y8u · 2023-11-23
> > >
> > > Thank you for the clarification and responses. Since my concerns on the dependence of models without details remain, I will keep my score.

---

> > > > ### Author Response · Authors · 2023-11-23
> > > > **Response to Reviewer 5Y8u**
> > > >
> > > > Thank you for reading through the response! Let us know if the evaluation results provided for the hierarchical entity performance and localization accuracy are enough from your point of view to address the highlighted weakness.
> > > >
> > > > And happy thanksgiving if you are celebrating!

---

### Official Review · Reviewer_W15R · 2023-11-01

**Soundness:** 4 excellent
**Presentation:** 2 fair
**Contribution:** 3 good
**Rating:** 6
**Confidence:** 4

**Summary:**

When applying LLM to Visually Rich Document (VRD) understanding, many methods use the two staged approaches: first execute the text recognition/serialization step and then execute the parsing step. However, lots of methods suffer from the need for large training data or are not able to predict hierarchical entities or hallucinations in domains other than text-only data. These problems are due to the absence of layout encoding within LLMs and the absence of a grounding mechanism ensuring the answer is not hallucinated. To overcome these challenges, the authors propose the five staged frameworks: OCR - chunking - prompt generation - LLM inference - decoding. The suggested framework is experimented with PaLM 2-S and compared to several publicly available baseline models on Visually Rich Document Understanding (VRDU) and Consolidated Receipt Dataset (CORD), resulting in a bigger performance margin than baseline methods.

**Strengths:**

- Suggest reasonable methods to tackle the challenges of visual document understanding.
- Provide rich information to reproduce experiments

**Weaknesses:**

* Though the suggested method seems agnostic to specific LLM, the authors experimented only with PaLM 2-s. To verify the superiority of the suggested framework, additional experiments using LLMs other than PaLM are needed (I think the additional experiment would enhance the presentation of the robustness of the proposed method).

**Questions:**

* In document representation, when generating prompts, how well do coordinate tokens work? Line-level segments with 2 coordinates are enough for various VRD data?
* Schema representation is important in the perspective of getting information in VRD. However, it would be vulnerable to hallucination. Does LLM properly parse JSON format?
* When doing Top-K sampling, we can choose Top-K sampling for individual N chunks and then merge, or do Top-K sampling for entire N chunks. I guess the latter method is better for the semantic integration quality (the similar reason that authors used the entire predicted tree value from a single LLM completion for hierarchical entities), but the authors used the former method. Is there a reason? I think the comparison may be interesting.

---

> ### Author Response · Authors · 2023-11-20
> **Response to Reviewer W15R (Part 1)**
>
> Firstly, we would like to thank reviewer W15R for taking the time to review our paper and providing great feedback! Please see our response below:
>
> > Though the suggested method seems agnostic to specific LLM, the authors experimented only with PaLM 2-s. To verify the superiority of the suggested framework, additional experiments using LLMs other than PaLM are needed (I think the additional experiment would enhance the presentation of the robustness of the proposed method).
>
> **[Response]** We agree with the feedback that evaluation numbers with other LLMs would better show the generality of our method. What we meant by LMDX being generic to the LLM is that it simply requires a simple text-in, text-out interface, which all LLMs provide (no extra input/output, or reliance on specific tokenization scheme). We will rephrase this in the next version of our paper to make this clearer.
>
> > In document representation, when generating prompts, how well do coordinate tokens work? Line-level segments with 2 coordinates are enough for various VRD data?
>
> **[Response]** Coordinate tokens are very important to a high quality extraction: this is showcased in the ablation study atTable 4, where we replace the coordinate tokens by line index (See appendix A.6 for how the prompts look). This allows us to not communicate any layout information to the model, while still using LMDX prompts and localizing the entities. Overall, coordinate tokens improves micro-F1 by an absolute 12-15% across data regimes on VRDU Ad-Buy Form Mixed.
>
> Line-level segments with 2 coordinates are indeed enough based on our experiments. We’ve tried with 4 coordinates, word-level and finer coordinate quantization buckets on an internal dataset which were all quality-neutral or negatives. We posit that this is due to the fact that those schemes add more coordinate tokens to the text sequence, which makes it further and further from the type of data the LLM was pretrained on. It becomes then harder for the LLM to correctly interpret the LMDX-style text sequences.

---

> > ### Author Response · Authors · 2023-11-20
> > **Response to Reviewer W15R (Part 2)**
> >
> > > Schema representation is important in the perspective of getting information in VRD. However, it would be vulnerable to hallucination. Does LLM properly parse JSON format?
> >
> > **[Response]** Yes, schema representation is indeed critical! In our experience, it takes a certain LLM size to properly parse the provided schema JSON. We initially started experimenting with PaLM 2-XXS, and the completions would include entity types not provided in the schema, and completely ignoring some entity types. Starting at PaLM 2-Small (the LLM we used), the model is able to follow the schema near perfectly. As part of this rebuttal, we’ve computed parse error rates on VRDU Ad-Buy Form Mixed, which shows the percentage of completions that were successfully parsed as json with the requested entities types present and values well formatted. See results below:
> >
> > | Dataset size | Invalid JSON | Invalid Entity Value Format | Entity Text Not Found |
> > |--------------|--------------|-----------------------------|-----------------------|
> > | 0            | 0.18%        | 0.04%                       | 0.59%                 |
> > | 10           | 0.27%        | 0.04%                       | 0.44%                 |
> > | 50           | 0.24%        | 0.00%                       | 0.17%                 |
> > | 100          | 0.24%        | 0.00%                       | 0.13%                 |
> > | 200          | 0.25%        | 0.00%                       | 0.09%                 |
> >
> > The different possible errors are:
> >
> > * Invalid JSON Formatting: This error refers to cases for which Python's json.loads(completion) fails on a LLM's completion. The JSON parsing error rate is below 0.3% in all training settings.
> > * Invalid Entity Value Format: This error refers to cases where the leaf entity value does not follow the expected ```text-segment-1 XX|YY text-segment-2 XX|YY``` format. The Invalid Entity Value Format Rate is below 0.05% in all training settings.
> > * Entity Text Not Found: This error refers to cases where the segment identifier is valid, but the entity text does not appear on the predicted segments. The Entity Text Not Found error rate is below 0.6% in all training settings.
> >
> > Despite those rare errors, with our sampling and decoding (aggregating) strategy, extraction is successful on every single document on the studied benchmarks.
> >
> > We will add those numbers to the appendix of the next version of our paper.
> >
> > > When doing Top-K sampling, we can choose Top-K sampling for individual N chunks and then merge, or do Top-K sampling for entire N chunks. I guess the latter method is better for the semantic integration quality (the similar reason that authors used the entire predicted tree value from a single LLM completion for hierarchical entities), but the authors used the former method. Is there a reason? I think the comparison may be interesting.
> >
> > **[Response]** We are not sure we fully understand the question, so please let us know if the following does not answer it. We think you are referring to the fact that we sample K completions from each chunk, do majority voting for each individual chunk, then merge predictions across all chunks, instead of doing majority voting across all N chunks directly.
> > Although that strategy sounds appealing, it would lead to a lower quality extraction. For instance, if we have a 3-chunk document, and we sample 2 completions per chunk, and the “registrant_name” entity only appears on one chunk, then we would have the following extraction from the LLM:
> >
> > ```
> > Chunk 1, Sample 1: { “registrant_name”: “John 10|20” }
> > Chunk 1, Sample 2: { “registrant_name”: “John 10|20” }
> >
> > Chunk 2, Sample 1: { “registrant_name”: null }
> > Chunk 2, Sample 2: { “registrant_name”: null }
> >
> > Chunk 3, Sample 1: { “registrant_name”: null }
> > Chunk 3, Sample 2: { “registrant_name”: null }
> > ```
> >
> > Voting across all chunks would mean that the chosen value for “registrant_name” is null (4 completions out of 6 = 67%). Instead, voting on an individual chunk basis and merging predictions across chunks (our current strategy) would choose “John” as the value for  “registrant_name” (2 completions out of 2 = 100%).

---

### Official Review · Reviewer_Vt8a · 2023-11-02

**Soundness:** 3 good
**Presentation:** 3 good
**Contribution:** 2 fair
**Rating:** 6
**Confidence:** 3

**Summary:**

This paper proposes LMDX — a mechanism for information extraction from documents leveraging off-the-shelf Optical Character Recognition service and LLM prompt engineering approach with PALM LLM for processing the extracted information.

**Strengths:**

Strengths:
* The paper shows potential of using LLMs for information extraction from documents
* Ablation studies are interesting and show the value of fine-tuning the PALM LLM for document information extraction

**Weaknesses:**

Weaknesses:

* This paper proposes a mechanism for information extraction from documents leveraging off-the-shelf Optical Character Recognition service and complicated LLM prompt engineering approach for processing the extracted information. The main underlying assumption driving the complexity of the prompt engineering approach is limited context length of LLMs. However, models like Claude 2 are capable of working with 100K token context windows. Additionally, methods like RoPE scaling and other context length expansion approaches allow to increase the context size for other LLMs including open-source models. As there are effective ways to address the context length limitation, the presented prompt engineering approach is a somewhat incremental engineering contribution, especially given its complexity. The fine-tuned model, however, is of interest.
* While the proposed approach outperforms other baselines on VRDU and CORD benchmarks, the performance advantage clearly comes from using a powerful LLM. It would be important to compare this method to OCR+long-context LLMs such as Claude 2.
* Another reasonable baseline with the potential to achieve high performance on these benchmarks is a multi-modal vision-text LLM, for example GPT-4. It has potential to work out of the box, without requiring fine-tuning, and significantly outperform other baselines.
* Code is not provided.
* Many unexplained abbreviations: e.g., IOB, NER. Readers would benefit from expanding these abbreviations the first time they are used.

**Questions:**

Questions:
* In-context learning is likely to significantly improve performance on this task, have you tried any experiments with in-context demonstrations?

**Details Of Ethics Concerns:**

no ethics concerns

---

> ### Author Response · Authors · 2023-11-20
> **Response to Reviewer Vt8a (Part 1)**
>
> We would like to thank reviewer Vt8a for taking the time to review the paper and give us insightful feedback! See our response below:
>
> > This paper proposes a mechanism for information extraction from documents leveraging off-the-shelf Optical Character Recognition service and complicated LLM prompt engineering approach for processing the extracted information. The main underlying assumption driving the complexity of the prompt engineering approach is limited context length of LLMs. However, models like Claude 2 are capable of working with 100K token context windows. Additionally, methods like RoPE scaling and other context length expansion approaches allow to increase the context size for other LLMs including open-source models. As there are effective ways to address the context length limitation, the presented prompt engineering approach is a somewhat incremental engineering contribution, especially given its complexity. The fine-tuned model, however, is of interest.
>
> **[Response]** To clarify, the complexity of the prompt engineering is not driven by a goal to reduce the number of tokens due to limited context length, but rather to communicate the layout information of the document to the model (necessary for a high-quality extraction on many visually-rich documents), and to derive the extracted entities’ bounding boxes (a major requirement in document information extraction which we want our method to handle). Both are achieved through the coordinate tokens contained in the prompt.
>
> The chunking stage in the pipeline (Section 2.3), which reduces the number of tokens sent in each prompt, is detailed so that ML practitioners can use the methodology on LLMs with context length limitations or on LLMs that can not use their full context well [1]. However, that stage can be entirely skipped for LLMs that supports long context and can use it effectively. We will ensure to clarify this in the next version of our paper.
>
> Nonetheless, we do make an effort to reduce the number of tokens used by our method whenever it is quality neutral so that it can be used by ML practitioners at a lower cost and faster inference speed, but this is not the primary goal.
>
> Overall, we hope this clarifies details around the complexity of the prompt!
>
> [1] Liu, Nelson F., Kevin Lin, John Hewitt, Ashwin Paranjape, Michele Bevilacqua, Fabio Petroni, and Percy Liang. "Lost in the middle: How language models use long contexts." arXiv preprint arXiv:2307.03172 (2023).

---

> > ### Author Response · Authors · 2023-11-20
> > **Response to Reviewer Vt8a (Part 2)**
> >
> > > While the proposed approach outperforms other baselines on VRDU and CORD benchmarks, the performance advantage clearly comes from using a powerful LLM. It would be important to compare this method to OCR+long-context LLMs such as Claude 2.
> >
> > **[Response]** We fully agree that the performance advantages are in part due to leveraging an LLM, but we argue that the performance advantages also come from the other parts of the LMDX methodology (prompting strategy, decoding, base training). For instance, the inclusion of the coordinate tokens led to an absolute 12-15% micro-F1 improvement over not using them on VRDU Ad-Buy form. The base extraction training led to a 7-39% micro-F1 improvement per the ablation study in Table 4.
> >
> > We hope this supports our claim that the performance advantages come not only from the LLM, but also the prompting strategy which communicates the layout of the document to the model as well. LMDX furthermore localize its predictions within the document.
> >
> > Per your suggestion, we’ve computed baseline numbers with a simple long context LLM baseline for VRDU and CORD. To do so, we've used GPT3.5 as none of the authors have access to Claude 2 at the moment. GPT-3.5 supports a long context of 16k tokens which is a long enough context for documents from the VRDU and CORD benchmarks. We prompt the LLM with the raw OCR text along with extraction instructions for the entity values as shown below:
> >
> > ```
> > {RAW_OCR_TEXT}
> >
> > Given the document, extract the text value of the entities included in the schema in json format.
> > - The extraction must respect the JSON schema.
> > - Only extract entities specified in the schema. Do not skip any entity types.
> > - The values must only include text found in the document.
> > - Use null or [] for missing entity types.
> > - Do not indent the json you produce.
> > - Examples of valid string value format: "$ 1234.50", "John Do", null.
> > - Examples of valid list value format: ["$ 1234.50", "John Do"], [].
> >
> > Schema: {"file_date": "", "foreign_principle_name": "", "registrant_name": "", "registration_num": "", "signer_name": "", "signer_title": ""}
> > ```json
> > ```
> >
> > See the results below:
> >
> > |       Dataset      /        Model          | GPT-3.5 | LMDX       |
> > |--------------------------------------------|---------|------------|
> > | VRDU - Registration Form - Seen Template   | 67.23%  | **73.81%** |
> > | VRDU - Registration Form - Mixed Template  | 63.86%  | **71.65%** |
> > | VRDU - Registration Form - Unseen Template | 67.49%  | **74.94%** |
> > | VRDU - Ad-Buy Form - Mixed Template        | 30.05%  | **39.74%** |
> > | VRDU - Ad-Buy Form - Unseen Template       | 29.84%  | **39.33%** |
> > | CORD                                       | 58.25%  | **67.47%** |
> >
> > Note that GPT-3.5 predictions are unlocalized. We will include those numbers in the next version of the paper.

---

> > > ### Author Response · Authors · 2023-11-20
> > > **Response to Reviewer Vt8a (Part 3)**
> > >
> > > > Another reasonable baseline with the potential to achieve high performance on these benchmarks is a multi-modal vision-text LLM, for example GPT-4. It has potential to work out of the box, without requiring fine-tuning, and significantly outperform other baselines.
> > >
> > > **[Response]** We agree that adding a vision-text LLM baseline is a great idea! Unfortunately, the gpt-4-vision-preview model API was released after the paper submission deadline and is currently rate limited to 100 requests per day. For context, to get numbers on the current benchmarks, we would need around 10000 requests. Instead, as part of this rebuttal, we've computed baseline numbers of a strong open-source vision-text LLM, LLaVA-v1.5-13B, which we prompt with the document image and an extraction instruction. Sample prompt:
> > >
> > > ```
> > > <DOCUMENT_IMAGE>
> > >
> > > Given the document, extract the text value of the entities included in the schema in json format.
> > > - The extraction must respect the JSON schema.
> > > - Only extract entities specified in the schema. Do not skip any entity types.
> > > - The values must only include text found in the document.
> > > - Use null or [] for missing entity types.
> > > - Do not indent the json you produce.
> > > - Examples of valid string value format: "$ 1234.50", "John Do", null.
> > > - Examples of valid list value format: ["$ 1234.50", "John Do"], [].
> > >
> > > Schema: {"file_date": "", "foreign_principle_name": "", "registrant_name": "", "registration_num": "", "signer_name": "", "signer_title": ""}
> > > ```json
> > > ```
> > >
> > > See the results below:
> > >
> > > |       Dataset      /        Model          | GPT-3.5 | LLaVA-v1.5-13B | LMDX       |
> > > |--------------------------------------------|---------|----------------|------------|
> > > | VRDU - Registration Form - Seen Template   | 67.23%  | 5.29%          | **73.81%** |
> > > | VRDU - Registration Form - Mixed Template  | 63.86%  | 5.00%          | **71.65%** |
> > > | VRDU - Registration Form - Unseen Template | 67.49%  | 5.05%          | **74.94%** |
> > > | VRDU - Ad-Buy Form - Mixed Template        | 30.05%  | 0.34%          | **39.74%** |
> > > | VRDU - Ad-Buy Form - Unseen Template       | 29.84%  | 0.38%          | **39.33%** |
> > > | CORD                                       | 58.25%  | 4.78%          | **67.47%** |
> > >
> > > In general, LLaVa does not perform well on document information extraction task, with numerous JSON errors, hallucinated entity values, and OCR errors. We also would like to point out that, unlike LMDX, LLaVA and other vision-text LLMs can not specify the entity location within the document, and as such does not meet all desired properties of a document information extraction system.
> > >
> > > > Code is not provided.
> > >
> > > **[Response]** While the code is not provided as we are not allowed to release PaLM2-derived models, we tried to describe the system in as much detail as possible to help with reproducibility on other models, with exact prompts and chunking/decoding algorithm provided in appendix A.1, A.2, A.3 and A.6.
> > >
> > > > Many unexplained abbreviations: e.g., IOB, NER. Readers would benefit from expanding these abbreviations the first time they are used.
> > >
> > > **[Response]** We will update the next version of the paper to define the acronyms that were missing (NER, IOB, XML). Thank you for pointing it out!
> > >
> > > > In-context learning is likely to significantly improve performance on this task, have you tried any experiments with in-context demonstrations?
> > >
> > > **[Response]** Yes, we do! See the results on the CORD benchmark in the table below, where we prompt the LMDX base extractor model with N documents and extraction from the CORD training data using the following methods:
> > >
> > > - Random N: randomly select N documents.
> > > - Nearest Neighbors: Use sentenceT5 embeddings for nearest neighbor retrieval.
> > >
> > > See results below:
> > >
> > > | N-Shot            | 0      | 1      | 3      | 5      | 10     |
> > > |-------------------|--------|--------|--------|--------|--------|
> > > | Random            | 67.47% | 74.96% | 84.88% | 86.47% | 87.26% |
> > > | Nearest Neighbors | 67.47% | 87.73% | 90.98% | 92.28% | 92.82% |
> > >
> > >
> > > Overall, few-shot helps as expected on CORD. There is a significant gain going from 0-shot to 1/3-shot, with smaller incremental returns as the number of shots increases. The nearest neighbors method has proven most effective, as it retrieves documents from similar templates, and is matching the performance of finetuning at N=10 examples. We will include and detail those results in the next version of our paper.
> > >
> > > Thank you again for reviewing our paper, and let us know if you have further questions! Thank you!

---

> > > > ### Comment · Reviewer_Vt8a · 2023-11-21
> > > > **Thank you for your thorough response**
> > > >
> > > > Dear Authors,
> > > >
> > > > Thank you very much for your excellent response and additional experiments! In particular, thank you for the clarification regarding the context length and highlighting the importance of your coordinate token approach (both through the explanation in your response and additional experiments on a vision-text LLM). Thank you for including the GPT-3.5 results as well -- they justify the complexity of the proposed method and alleviate my concerns about the potential to achieve a similar level of performance out-of-the-box. I also welcome your points about limited accessibility of Claude and GPT-4 and I agree that your experiments provide an excellent alternative. In light of the new experiments and clarifications, I am happy to recommend acceptance and raise my score.
> > > >
> > > > Thank you!

---

> > > > > ### Author Response · Authors · 2023-11-23
> > > > > **Response to Reviewer Vt8a**
> > > > >
> > > > > Thank you for the kind words, and the updated score!

---

### Author Response · Authors · 2023-11-23
**General response - changes made to the paper (part 1)**

We would like to thank all reviewers for their thoughtful comments and feedback, we do feel it improved the paper substantially.

### Context

To provide additional context on the paper, information extraction systems for visually-rich documents have the following desired properties:

* High-quality extraction at low data: as there are a quasi-infinite amount of templates and document types, most parsers are built with very little amount of training data, hence reaching high quality in the low data regime is critical.
* Hierarchical entity support: apart from simply extracting entities within the document, a document information extraction system should group entities within logical groups (e.g. items and prices should be associated with one another in an invoice or a receipt). Without it, the extractions are not as useful.
* Entity localization: Since document information extractors are usually integrated with human-in-the-loop in document processing workflows, they should specify the location of the entities within the document. This allows fast verification of the their predictions for reliability-critical applications like Know-Your-Customer, tax documents and medical records processing (e.g. human operators can directly verify that the highlighted entity is correct, instead of sifting through multiple document pages to verify it).

Because of those requirements, recent works still rely on fully custom architectures in 2023 [1-7], hence the field has not benefited from the massive improvements that LLMs have brought to most natural language processing (NLP) tasks. To the best of our knowledge, prior to this paper, there has not been work successfully applying LLMs for information extraction and localization on visually-rich documents. What we hope we demonstrated in this paper is that, with appropriate prompting and decoding, it is possible to use existing LLMs and meet all desired properties of entity extraction systems. We achieve this through the introduction of coordinate tokens, which allows encoding the layout modality (needed for a high quality extraction), localizing the predicted entities within the document and discarding all hallucinated entity values (a common problem with LLMs). Overall, the results strongly support that LMDX is effective.

We hope this additional background helps contextualize the paper better within the broader field!

[1] Wang, Dongsheng, et al. "DocGraphLM: Documental Graph Language Model for Information Extraction." Proceedings of the 46th International ACM SIGIR Conference on Research and Development in Information Retrieval. 2023.

[2] Lee, Chen-Yu, et al. "FormNetV2: Multimodal Graph Contrastive Learning for Form Document Information Extraction." arXiv preprint arXiv:2305.02549 (2023).

[3] Tang, Zineng, et al. "Unifying vision, text, and layout for universal document processing." Proceedings of the IEEE/CVF Conference on Computer Vision and Pattern Recognition. 2023.

[4] Appalaraju, Srikar, et al. "DocFormerv2: Local Features for Document Understanding." arXiv preprint arXiv:2306.01733 (2023).

[5] Chen, Jiayi, et al. "On Task-personalized Multimodal Few-shot Learning for Visually-rich Document Entity Retrieval." arXiv preprint arXiv:2311.00693 (2023).

[6] Shi, Dengliang, et al. "LayoutGCN: A Lightweight Architecture for Visually Rich Document Understanding." International Conference on Document Analysis and Recognition. Cham: Springer Nature Switzerland, 2023.

[7] Liao, Haofu, et al. "DocTr: Document transformer for structured information extraction in documents." Proceedings of the IEEE/CVF International Conference on Computer Vision. 2023.

---

> ### Author Response · Authors · 2023-11-23
> **General response - changes made to the paper (part 2)**
>
> ### Changes made to the paper
>
> Following the received feedback, we've done the following changes to the paper:
>
> * **(reviewer 5Y8u)** We've added an evaluation and discussion of the localization accuracy in Table 2 and Section 3.3 (page 7-8). This allows to compare LMDX's localization performance versus baselines, independently of the extraction quality.
> * **(reviewer 5Y8u)** We've clarified that Line Item F1 in Table 2 and Section 3.3 (page 7) is the evaluation of hierarchical entity performance, and independently compares how LMDX F1 performance on hierarchical entities compares to baselines.
> * **(reviewer Vt8a)** We've added large models baselines to the results Tables 2 and 3 (section 3.3, page 7-8). Those baselines are using GPT-3.5 and LLaVA-v1.5-13B and are fully detailed in Appendix A.7. Unlike LMDX, those baselines do not allow localizing the entities, and as such do not meet all desired properties of document information extraction systems.
> * **(reviewer Vt8a)** We've added section 3.5 (page 8-9) which contains in-context learning results on CORD with multiple retrieval methods, and discussion of those results. Overall, nearest neighbors work the best, as it retrieves documents from the same template as the target document (see Appendix A.10 for a more detailed discussion and sample retrievals).
> * **(reviewer W15R)** We've added discussion on sampling error rates in Appendix A.9, detailing the different errors that can happen during decoding of the LLM's completions. Errors are very low overall, and with our sampling strategy, all documents on our benchmarks have a successful extraction.
> * **(reviewer Vt8a)** We've clarified the role of chunking in section 2.3 (page 3) with regards to long-context LLMs.
> * **(reviewer 5Y8u)** We've clarified in section 3.2 (page 6) that we use the benchmark's provided OCRs for all our experiments and for all models and baselines taking text as input, leading to a fair comparison between them.
> * **(reviewer W15R)** We've clarified in the contributions in Section 1 (page 2) that LMDX only requires a simple text-input/text-output interface that all LLMs provide, and does not rely on other details such as a particular tokenization.
> * **(reviewer Vt8a)** We've defined the unexplained abbreviations (IOB, NER) before using them.
> * Finally, we've done general rephrasing and small changes to make everything fit within the page limit. In particular, Figure 3 (Diagram about the LMDX training phases) was removed as it did not provide any additional information from what was contained in the text content.
>
> We would like to thank you for taking the time to provide insights and suggestions on our paper. We do believe it improved our paper significantly! Please let us know if any question/concern remains unanswered, and happy Thanksgiving if you're celebrating it!